# Predictive Feature Caching for Training-free Acceleration of Molecular Geometry Generation

**Johanna Sommer**[1,2]**, Nils Fleischmann**[1]**, John Rachwan**[1]**,**
**Stephan Günnemann**[1,2]**, Bertrand Charpentier**[1]

[1] Pruna AI
[2] Technical University of Munich, Germany

`{firstname.lastname}@pruna.ai`

**Reviewed on OpenReview:** `https://openreview.net/forum?id=NaLVutCHCI`

## Abstract

Flow matching models generate high-fidelity molecular geometries but incur significant computational costs during inference, requiring hundreds of neural network evaluations. This inference cost becomes the primary bottleneck when such models are employed in practice to sample large numbers of molecular candidates. This work presents a training-free caching strategy that accelerates molecular geometry generation by predicting intermediate hidden states across solver steps. This caching scheme operates directly on the SE(3)-equivariant backbone, is compatible with pretrained models, and is orthogonal to existing training-based accelerations and system-level optimizations. Experiments on molecular geometry generation demonstrate that caching achieves a twofold reduction in wall-clock inference time at matched sample quality and a speedup of up to $3\times$ with minimal sample quality degradation. Because these gains compound with other optimizations, applying caching alongside other general, lossless optimizations yield as much as a $7\times$ speedup. We make the code available at: `https://github.com/PrunaAI/caching-for-molecule-generation`

## 1 Introduction

Deep learning, particularly deep generative modeling, is rapidly transforming molecular design by enabling the de-novo creation of molecular geometries (Wang et al., 2025; Alakhdar et al., 2024; Tang et al., 2024). Among generative methods, flow matching models have emerged as the state-of-the-art for generating high-quality molecular geometries. Their iterative denoising nature allows for flexible modeling of complex geometric distributions. While earlier approaches focused on SMILES strings or molecular graphs, the direct generation of molecular geometries, represented as featurized point clouds, has gained traction due to its fidelity in capturing geometric and physicochemical constraints critical for real-world efficacy.

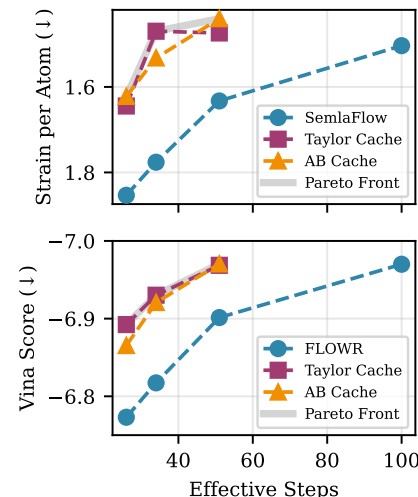

Figure 1: Caching defines the speed-quality Pareto front.

Traditional drug discovery pipelines rely on combinatorial screening of known compounds. In contrast, generative models aim to sample directly from the underlying chemical distribution. This approach offers a more controlled way to navigate the chemical space than, for example, virtual or high-throughput screening (Johansson et al., 2024). However, in practice, these models are still required to sample 500,000 or even over one million compounds (Shen et al., 2024; Koziarski et al., 2024). Consequently, the molecular generator's inference time becomes the dominant bottleneck. This holds especially for diffusion or flow matching models, whose sampling can require hundreds of neural network evaluations for a single molecule, resulting in prohibitively slow sampling at scale.

All practical acceleration methods that have been introduced specifically for molecular generation *require additional training*, incurring data, compute, and time overhead. Trajectory reparameterization trains diffusion models to straighten their stochastic paths, thereby reducing the number of steps required to reach data-like samples (Ni et al., 2025). Progressive distillation trains a student to replace several teacher denoising steps with one, iteratively halving the sampling budget (Lacombe and Vaidya, 2024). Latent methods train an autoencoder and then a generator in the compressed space, allowing denoising to run in a lower-dimensional latent space and cutting per-step computation (Xu et al., 2023).

While architectural refinements and diffusion process adjustments have led to significant gains in the speed and efficiency of molecular generative models, we pursue a *complementary* and *training-free* direction. Inspired by recent advances in image generation, we accelerate molecular geometry generation by skipping redundant network computations and predicting intermediate network activations from previously computed ones during sampling. By leveraging previously computed features across time steps, our method reduces redundant computation, achieving a $2\times$ speedup at matched sample quality and a $3\times$ speedup in generation with marginal impact on generation quality. We also find that caching can serve as a refinement mechanism, improving sample quality relative to the base model at the same effective inference budget. As shown in Figure 1, caching defines the Pareto front of the speed-quality tradeoff compared to the base model at the full inference budget.

Our contributions are as follows:

- We investigate predictive feature caching as the first training-free scheme that accelerates molecular geometry generation models at little to no quality loss. The method is drop-in for pretrained models and reduces inference cost while preserving generation quality.

- We transfer and analyze multiple feature-caching strategies to unconditional and structure-based molecular geometry generation, operating directly on molecular geometries and SE(3)-equivariant representations.

- We show that our approach is complementary to general post-training optimizations, highlighting that they can be combined for significant inference-time speedups.

## 2 Related Work

**Diffusion caching.** Caching was first introduced for image diffusion models, where prior work (Wimbauer et al., 2023; Ma et al., 2023; Li et al., 2024) observed temporal redundancy in the U-Net's high-level features, which can be exploited by caching and reusing them across successive denoising steps. Extending this to diffusion transformers (DiTs), Selvaraju et al. (2024) report analogous temporal similarity in attention and MLP activations and propose reusing them over multiple steps. Chen et al. (2024) leverages role asymmetries across blocks, caching rear blocks early and front blocks late via a DiT-specific $\Delta$-cache. Rather than reusing features from the most recent step, TaylorSeer (Liu et al., 2025a) predicts future features via a Taylor-series expansion, while AB-Cache (Yu et al., 2025) employs an Adams–Bashforth scheme to compute a predictions based on previously computed features. Classical caching recomputes features at a predefined interval; in contrast, TeaCache adaptively decides when to refresh the cache based on the inputs of the DiT (Liu et al., 2025b). In image generation, Region Adaptive Sampling allocates computation spatially, updating only regions in focus while reusing cached features elsewhere (Liu et al., 2025c). (Choi et al., 2025) introduces a learned caching framework that selectively applies lightweight linear modulation to cached activations. For video generation models, caching must respect temporal coherence and inter-frame redundancy, hence various caching strategies specifically tailored to video generation have been proposed (Sun et al., 2025; Lv et al., 2025; Yuan et al., 2024; Liu et al., 2025b; Ma et al., 2025). Previous work has primarily focused on various caching approaches for the image and video domains; none of which have been transferred to deep generative models in the molecular domain.

**De-novo molecular geometry generation.** Work on de-novo molecule generation today focuses on directly generating molecules as their 3D coordinates, representing structures in continuous Euclidean space, and parameterizing them as Cartesian or internal coordinates alongside atom types. Within this line of work, variational autoencoders learn a latent space over geometries and decode molecules with equivariant architectures that enforce basic geometric constraints (Ragoza et al., 2020). Building on that, autoregressive models place atoms or fragments sequentially in a 3D space, conditioned on the growing partial structure (Gebauer et al., 2020; Luo and Ji, 2022). In parallel, normalizing flow-based methods define invertible transformations over coordinates to provide exact likelihoods under $E(n)$ / $SE(3)$ equivariance (Satorras et al., 2022). Most recently, diffusion-based approaches have become prominent: some follow classical score-based diffusion (Hoogeboom et al., 2022; Huang et al., 2022; 2023; Vignac et al., 2023; Qiang et al., 2023; Morehead and Cheng, 2024; Wu et al., 2022; Xu et al., 2023; Reidenbach et al., 2025; Hong et al., 2025; Feng et al., 2025; Ni et al., 2025; Irwin et al., 2025) while others adopt the closely related flow-matching formulation to learn continuous probability flows (Song et al., 2023; Dunn and Koes, 2024; Joshi et al., 2025; Dunn and Koes, 2025). Although such models generate molecules unconditionally, they can be adapted for downstream tasks, such as property optimization or shape-constrained generation, via, for example, diffusion guidance at inference time (Ayadi et al., 2025; Ketata et al., 2024). In contrast, several architectures explicitly incorporate the protein as conditioning during training to directly model protein-ligand interactions (Guan et al., 2023; Schneuing et al., 2024; Cremer et al., 2024; Schneuing et al., 2025).

## 3 Flow Matching for Molecular Geometry Generation

Let $p_{\text{data}}$ denote a target data distribution on a state space $\mathcal{X}$, and let $p_{\text{noise}}$ be a simple base distribution on $\mathcal{X}$. Flow matching transports $p_{\text{noise}}$ at $t = 0$ to $p_{\text{data}}$ at $t = 1$ by learning a time-dependent vector field $\{v_\theta(\cdot, t)\}_{t \in [0,1]}$ (Lipman et al., 2023). The field $v_\theta$ induces a flow $\Phi_{s \to t}$ whose pushforward maps a probability path $(p_t)_{t \in [0,1]}$ from $p_0 = p_{\text{noise}}$ to $p_1 = p_{\text{data}}$ under the dynamics $\dot{x}_t = v_\theta(x_t, t)$.

Conditional flow matching (CFM) allows training the vector field by specifying, for each data sample $x_1 \sim p_{\text{data}}$, a conditional path distribution $\{p_{t|1}(\cdot \mid x_1)\}_{t \in [0,1]}$ and its conditional velocity field $u_t(x_t \mid x_1) \in T_{x_t}\mathcal{X}$, where $T_{x_t}\mathcal{X}$ denotes the tangent space of $\mathcal{X}$ at $x_t$ (Tong et al., 2024). For unconditional generation, we condition only on $x_1$, but other conditioning choices are possible, for example, an auxiliary variable defining a linear-interpolation bridge (Liu et al., 2022). The training objective learns the velocity field by regressing

$$v_\theta(x_t, t) \approx u_t(x_t \mid x_1). \tag{1}$$

During sampling, we integrate this ODE on a discrete time grid $0 = t_0 < t_1 < \cdots < t_K = 1$. The first-order Euler scheme

$$x_{k+1} = x_k + \Delta t_k\, v_\theta(x_k, t_k), \qquad \Delta t_k := t_{k+1} - t_k, \tag{2}$$

approximates the transformation from the base to the data distribution in $K$ discrete steps.

**Molecular geometry parameterization.** Molecular geometries comprise multiple atoms, each with 3D coordinates and an atom type, as well as bonds between atoms labeled by discrete bond orders. We model a molecule as a tuple

$$x = (c, a, b) \in \mathcal{X} := \underbrace{\mathbb{R}^{N \times 3}}_{\text{coords.}} \times \underbrace{\mathcal{A}^N}_{\text{atom types}} \times \underbrace{\mathcal{B}^{\mathcal{E}}}_{\text{bond orders}}, \tag{3}$$

and we learn the joint distribution $p(x)$. These variables are regressed jointly, yielding a single parameterization that induces a coupled vector field on coordinates, atom types, and bond orders. Given an atom count $n$, $x_0 \sim p_{\text{noise}}(\cdot \mid n)$ is drawn and integrated from $t = 0$ to $1$ as stated in Equation 2 to obtain joint samples $x = (c, a, b)$.

**Equivariance.** Molecular geometries are unchanged by global rotations and translations, so enforcing $E(3)$ equivariance prevents the model from learning spurious patterns. Let $G$ act on the state space $\mathcal{X}$ via a representation $\rho_{\mathcal{X}} : G \to \text{GL}(\mathcal{X})$, where $\text{GL}(\mathcal{X})$ denotes the group of invertible linear maps on $\mathcal{X}$. Here, $\rho_{\mathcal{X}}(g)\, x$ denotes the configuration obtained by applying $g$ to $x$, for example by rotating or translating coordinates and permuting atoms. We denote the pushforward of this action on tangent vectors by $d\rho_{\mathcal{X}}$.

A density $p$ on $\mathcal{X}$ is $G$-invariant if $p(\rho_{\mathcal{X}}(g)\,x) = p(x)$ for all $g \in G$. A function $f : \mathcal{X} \to T\mathcal{X}$ is $G$-equivariant with respect to $(\rho_{\mathcal{X}}, d\rho_{\mathcal{X}})$ if

$$d\rho_{\mathcal{X}}(g)\,f(x) = f(\rho_{\mathcal{X}}(g)\,x) \quad \forall g \in G. \tag{4}$$

We enforce $E(3) \times S_N$ equivariance of the velocity field $v_\theta(\cdot, t)$. If the base density $p_0$ is $G$-invariant and $v_\theta(\cdot, t)$ is $G$-equivariant for all $t$, then the terminal density at $t = 1$ is $G$-invariant. For molecules we take $G = E(3) \times S_N$: the Euclidean group in 3D acting on coordinates and the symmetric group on $N$ atoms acting by permutation on atoms. This is enforced by using isotropic coordinate noise and $G$-equivariant updates.

## 4 Predictive Feature Caching for the Molecular Domain

Evaluating the time-dependent vector field $v_\theta(x_t, t)$ dominates the inference cost of flow matching. The ODE solver has to query $v_\theta$ many times at closely spaced time steps, and each query runs the full backbone, i.e. the main neural network that parameterizes $v_\theta$, on inputs that change smoothly with $t$. As a result, intermediate activations at each network block evolve along a smooth feature trajectory over time.

In this work, we investigate *predictive feature caching* as a means to significantly reduce inference-time computational overhead by leveraging smooth feature trajectories during the generation of molecular geometries. Instead of recomputing similar features from scratch at every solver step, we store features at selected "checkpoint" times. We then reuse or predict features at nearby times to avoid full forward passes.

Recall from section 3 that we sample by integrating $\dot{x}_t = v_\theta(x_t, t)$ with $x_t = (c, a, b)$, where $v_\theta$ is implemented by a shared backbone. Let the backbone be a composition of blocks $F^L \circ \cdots \circ F^1$. At solver time $t$, we denote the input to block $l$ (of $L$ blocks in total) by $x_t^l$ and the block's output by $x_t^{l+1} := F^l(x_t^l)$. Because $x_t$ evolves under an ODE with a smooth right-hand side and the network is continuous in $(x, t)$, $x_t^l$ will vary smoothly with $t$ as shown in Figure 2. This smoothness provides a regularity that predictive caching exploits.

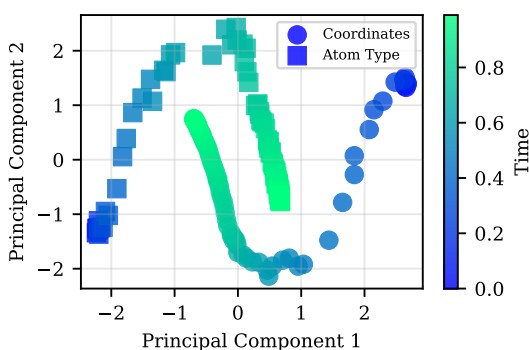

Figure 2: Projection onto the first two principal components of a single molecule's generation trajectory. Both coordinates and atom types evolve smoothly over solver steps.

Motivated by the strictly sequential information flow in transformer-based architectures for flow matching models and the high predictability of late-layer features, we follow Guan et al. (2025) and adopt a *last-block forecast*: at each time step, we apply the predictor only to the last block $L$, which avoids recomputation of the entire prefix $F^{L-1} \circ \cdots \circ F^1$. For notational convenience, we denote the last block as $F := F^L$ and its input feature as $h_t := x_t^L$.

At each cached time step $t_k$, we evaluate the backbone $F$. Within a caching interval of size $D$ starting at $t_k$, we substitute the backbone evaluation $F(h_{t_{k+r}})$ with a cheap function call $\widehat{F}(h_{t_{k+r}})$ satisfying $\widehat{F}(h_{t_{k+r}}) \approx F(h_{t_{k+r}})$ for the subsequent steps $r \in \{1, \ldots, D\}$. By utilizing $\widehat{F}$ in place of the full backbone, we can bypass the primary inference bottleneck. Naive caching simply reuses the last computed feature without forecasting, such that $\widehat{F}(h_{t_{k+r}}) = F(h_{t_k})$ for all $r \in \{1, \ldots, D\}$.

**TaylorSeer predictive caching.** Naive caching is cheap but accumulates staleness error because the features drift as $t$ advances. Liu et al. (2025a) address this with TaylorSeer *predictive caching*: it leverages local Taylor expansions of the feature trajectory to forecast intermediate features. At predefined time steps spaced every $D$ solver steps, we perform a standard forward pass and materialize the cache

$$C(h_t) = \left\{ F(h_t),\, \Delta F(h_t),\, \ldots,\, \Delta^m F(h_t) \right\}. \tag{5}$$

The $m$th-order TaylorSeer predictor forecasts features at time $t_{k+r}$ using a local Taylor expansion, where derivatives are approximated via finite differences $\Delta^i F(h_{t_k})$:

$$\widehat{F}_{\text{TS}(m)}(h_{t_{k+r}}) = F(h_{t_k}) + \sum_{i=1}^{m} \frac{\Delta^i F(h_{t_k})}{i!\, D^i}(-r)^i. \tag{6}$$

For $m = 0$, this reduces to naive caching. Concretely, every $D$ solver steps, we obtain $F(h_{t_k})$ and populate the cache $C(h_{t_k})$. For any time step $t_{k+r}$ within the current window $\{t_k, \dots, t_{k+D}\}$, we then forecast $\widehat{F}_{\text{TS}(m)}(h_{t_{k+r}})$. In our implementation, we guarantee, regardless of the caching interval $D$, that the last inference step is computed via $F(h_T)$.

**Adams–Bashforth caching.** Yu et al. (2025) similarly argues that the smooth feature trajectory can be exploited by applying a $j$-step Adams–Bashforth (AB) linear multistep forecast. For flow matching, this yields the $j$th-order linear recursion

$$\widehat{F}_{\text{AB}(j)}(h_{t_{k+r}}) := \sum_{i=1}^{j} (-1)^{i+1} \binom{j}{i} F(h_{t_{k+r-i}}), \tag{7}$$

which uses the last $j$ cached outputs to predict the current output. The implementation is similar to that of TaylorSeer caching; every $D$ solver steps, we populate the cache of backbone outputs, predict subsequent steps with $\widehat{F}_{\text{AB}(j)}(h_{t_{k+r}})$, and ensure that the last step is computed with $F(h_T)$.

**Equivariance.** Caching, as we have established it, is a time-scalar linear combination of cached features and finite differences. These operations commute with the $G$ action. Using the Euler update, equivariance yields the commutation relation

$$\rho_{\mathcal{X}}(g)\, x_{k+1} = \rho_{\mathcal{X}}(g)\, x_k + \Delta t_k \, d\rho_{\mathcal{X}}(g)\, v_\theta(x_k, t_k) = x'_{k+1} \quad \text{with} \quad x'_k = \rho_{\mathcal{X}}(g)\, x_k, \tag{8}$$

so each discrete step preserves the symmetry action. Consequently, if $v_\theta(\cdot, t)$ is $G$-equivariant at cached time steps and the base density $p_{\text{noise}}$ is $G$-invariant (see Sec. 3), the forecasted evaluations are $G$-equivariant throughout sampling, and the terminal density remains $G$-invariant. As a result, the discussed predictive feature caching preserves equivariance of the generation process.

**Limitations.** Feature caching relies on trajectory smoothness, a property we demonstrate empirically across multiple molecule generation architectures. However, specific architectural design choices or complex learned dynamics might induce non-smooth trajectories, for which a fixed caching interval may be insufficient to accurately approximate high-frequency feature variations. Such transient regions motivate future research to dynamically adjust the caching frequency based on local trajectory complexity.

## 5 Experiments

We evaluate caching on equivariant flow-matching generators for the molecular geometries. The primary objective is to characterize the quality–speed trade-off of two caching variants, Taylor forecasting and the Adams–Bashforth (AB) multi-step method, for both unconditional and conditional molecular geometry generation. We do so by evaluating inference overhead alongside standard quality metrics. Crucially, we aim to demonstrate that caching enables significant acceleration that remains largely lossless relative to the base model. Furthermore, we show that caching strategies consistently outperform the common practice of reducing inference steps (i.e., uniform step reduction) at lower step budgets, thereby forming the efficiency frontier. We also find that caching can act not only as an acceleration technique, but as a refinement mechanism improving sample quality. Finally, we examine how caching composes with orthogonal post-training optimizations.

Table 1: Comparison of the SemlaFlow, Tabasco, FLOWR and FLOWR.root base models and their cached variants. We highlight in **bold** results that are better than or equal to the results of the base model at full inference steps.

| | | D | Mode | Valid (PRC) ↑ | Energy ↓ | Energy p.A ↓ | Strain ↓ | Strain p.A ↓ | Opt. RMSD ↓ | Lipinski ↑ | Throughput ↑ |
|---|---|---|---|---|---|---|---|---|---|---|---|
| GEOM | Semla Flow | - | Base 100 | 0.88 ± 0.01 | 108.8 ± 0.9 | 2.38 ± 0.01 | 69.6 ± 0.7 | 1.50 ± 0.01 | 0.86 ± 0.00 | 4.82 ± 0.00 | 11.4 ± 0.1 |
| | | - | Base 51 | 0.86 ± 0.01 | 115.5 ± 0.8 | 2.51 ± 0.01 | 75.9 ± 0.8 | 1.63 ± 0.01 | 0.88 ± 0.00 | **4.82 ± 0.00** | **21.9 ± 0.2** |
| | | 2 | Cache (TS) | 0.85 ± 0.00 | **103.1 ± 1.2** | **2.28 ± 0.02** | **67.5 ± 1.0** | **1.48 ± 0.01** | 0.87 ± 0.00 | **4.82 ± 0.00** | 21.8 ± 0.2 |
| | | | Cache (AB) | 0.87 ± 0.00 | **96.5 ± 0.9** | **2.15 ± 0.01** | **62.8 ± 0.6** | **1.40 ± 0.01** | 0.87 ± 0.01 | 4.79 ± 0.00 | **22.1 ± 0.0** |
| | Tabasco (hot) | - | Base 50 | 0.90 ± 0.00 | 70.5 ± 0.2 | 2.80 ± 0.01 | 39.9 ± 0.1 | 1.52 ± 0.01 | 0.83 ± 0.01 | 4.93 ± 0.00 | 266.4 ± 2.0 |
| | | - | Base 26 | 0.88 ± 0.00 | 75.2 ± 0.5 | 2.96 ± 0.03 | 42.6 ± 0.2 | 1.63 ± 0.01 | 0.92 ± 0.01 | **4.92 ± 0.00** | **508.4 ± 2.6** |
| | | 2 | Cache (TS) | **0.90 ± 0.00** | 75.5 ± 0.5 | 2.99 ± 0.01 | 41.0 ± 0.8 | 1.58 ± 0.03 | **0.79 ± 0.00** | **4.92 ± 0.00** | 446.8 ± 5.8 |
| | | | Cache (AB) | **0.90 ± 0.00** | 73.2 ± 0.4 | 2.91 ± 0.02 | **39.8 ± 0.2** | **1.52 ± 0.01** | **0.81 ± 0.00** | 4.91 ± 0.00 | 453.4 ± 5.6 |

| | | D | Mode | Vina ↓ | VinaMin ↓ | Strain ↓ | PB Validity ↑ | Angle W1 ↓ | Length W1 ↓ | Dihedral W1 ↓ | Throughput ↑ |
|---|---|---|---|---|---|---|---|---|---|---|---|
| SPINDR | FLOWR | - | Base 100 | -6.97 ± 0.91 | -7.27 ± 0.89 | 90.2 ± 52.2 | 0.94 ± 0.19 | 1.079 | 0.00473 | 3.836 | 6.76 ± 2.81 |
| | | - | Base 51 | -6.90 ± 0.93 | -7.21 ± 0.92 | 99.6 ± 57.2 | 0.93 ± 0.22 | 1.187 | 0.00622 | 4.152 | **11.58 ± 4.50** |
| | | 2 | Cache (TS) | **-6.97 ± 0.93** | **-7.27 ± 0.91** | 93.5 ± 55.3 | **0.94 ± 0.20** | 1.129 | 0.00782 | 3.917 | 11.38 ± 4.33 |
| | | | Cache (AB) | **-6.97 ± 0.92** | **-7.28 ± 0.90** | 93.2 ± 55.2 | **0.94 ± 0.20** | 1.124 | **0.00458** | 3.893 | 11.46 ± 4.40 |
| | FLOWR root | - | Base 100 | -7.42 ± 0.82 | -7.61 ± 0.85 | 47.4 ± 34.7 | 0.99 ± 0.07 | 0.313 | 0.00826 | 3.285 | 4.75 ± 1.88 |
| | | - | Base 51 | -7.37 ± 0.83 | -7.58 ± 0.84 | 49.5 ± 35.9 | 0.97 ± 0.11 | 0.336 | **0.00817** | 3.691 | **7.90 ± 2.88** |
| | | 2 | Cache (TS) | -7.38 ± 0.82 | -7.59 ± 0.83 | 47.5 ± 35.0 | **0.99 ± 0.07** | **0.294** | 0.00825 | 3.475 | **7.93 ± 2.82** |
| | | | Cache (AB) | -7.39 ± 0.82 | **-7.61 ± 0.83** | **47.4 ± 35.1** | **0.99 ± 0.08** | **0.298** | 0.00833 | **3.172** | 7.76 ± 2.84 |
| Crossdock. | FLOWR root | - | Base 100 | -7.52 ± 0.54 | -8.00 ± 0.44 | 37.92 ± 28.48 | 0.92 ± 0.18 | 1.415 | 0.01965 | 3.536 | 3.04 ± 1.08 |
| | | - | Base 51 | -7.45 ± 0.52 | -7.96 ± 0.43 | 41.34 ± 30.58 | 0.91 ± 0.22 | **1.362** | 0.02011 | 3.767 | **5.24 ± 1.76** |
| | | 2 | Cache (TS) | -7.49 ± 0.53 | -7.98 ± 0.43 | 39.72 ± 29.55 | 0.91 ± 0.20 | **1.407** | 0.02009 | 3.613 | **5.19 ± 1.72** |
| | | | Cache (AB) | **-7.52 ± 0.52** | **-8.00 ± 0.43** | 40.04 ± 29.64 | **0.92 ± 0.20** | 1.504 | **0.00452** | **3.501** | 5.21 ± 1.33 |

**Evaluation.** For unconditional generation, the GEOM Drugs dataset (Axelrod and Gomez-Bombarelli, 2022), which contains 1 million high-quality conformers of drug-like molecules, is used to assess model performance as an unconditional molecular generator. Data splits and preprocessing follow the guidance given in Vignac et al. (2023); Le et al. (2023). To evaluate conditional generation in the form of structure-based geometry generation, the SPINDR dataset, which contains ligand-pocket co-crystal complexes, is employed as described in Cremer et al. (2025), as well as the Crossdocked2020 dataset, a large-scale collection of approximately 22.5 million protein-ligand binding poses designed for structure-based machine learning (Francoeur et al., 2020).

As a base models, we use SemlaFlow (Irwin et al., 2025), Tabasco (Vonessen et al., 2025), FLOWR (Cremer et al., 2025) and FLOWR.root (Cremer et al., 2026), and use the pretrained weights provided by the authors for all datasets. Unless explicitly stated otherwise, we use the default hyperparameters reported in the respective papers. For the Tabasco model, we choose 50 steps as the base configuration, as ablations in Vonessen et al. (2025) demonstrate that "additional steps have no effect on molecular quality", see Appendix A. All metrics presented in the subsequent experiments are calculated by sampling from the distribution of molecule sizes in the test set, followed by generating molecules with the sampled number of atoms through integration of the trained ODE. For conditional geometry generation, 10,000 molecules are sampled over three random seeds, while for structure-based generation, 100 ligands are sampled per pocket in the test set. All experiments are conducted on a single NVIDIA H100 PCIe GPU.

For unconditional molecular geometry generation, the objective is to learn the distribution of stable, drug-like molecules and generate novel three-dimensional structures that are physically realistic and synthetically feasible. In structure-based drug design, the goal is to generate ligands that exhibit strong structural and chemical complementarity to a given protein binding site. Model performance is further assessed using predicted binding affinities, such as VINA scores, to ensure that generated molecules are not only valid but also potentially effective for targeted therapeutic applications. We closely follow the respective work's evaluation protocol and detail the quality metrics used in the subsequent experiments in Appendix C. Lastly, to measure inference time, we report throughput as the number of molecules sampled per second.

Table 2: Comparison of the SemlaFlow, Tabasco, FLOWR and FLOWR.root base models and their cached variants. We highlight in **bold** the best results per effective inference steps.

| | | D | Mode | Valid (PRC) ↑ | Energy ↓ | Energy p.A ↓ | Strain ↓ | Strain p.A ↓ | Opt. RMSD ↓ | Lipinski ↑ |
|---|---|---|---|---|---|---|---|---|---|---|
| GEOM | SemlaFlow | - | Base 34 | **0.85 ± 0.00** | 120.3 ± 1.6 | 2.62 ± 0.03 | 82.0 ± 1.1 | 1.78 ± 0.02 | 0.90 ± 0.01 | **4.80 ± 0.00** |
| | | 3 | Cache (TS) | **0.85 ± 0.01** | 103.8 ± 0.5 | 2.31 ± 0.01 | 70.0 ± 0.5 | 1.56 ± 0.01 | **0.89 ± 0.00** | 4.78 ± 0.00 |
| | | | Cache (AB) | **0.85 ± 0.00** | **100.5 ± 1.0** | **2.25 ± 0.02** | **67.7 ± 0.7** | **1.51 ± 0.02** | 0.90 ± 0.01 | **4.80 ± 0.00** |
| | | - | Base 26 | **0.82 ± 0.00** | 123.5 ± 0.4 | 2.69 ± 0.01 | 85.5 ± 0.5 | 1.85 ± 0.01 | 0.92 ± 0.00 | **4.79 ± 0.00** |
| | | 4 | Cache (TS) | **0.82 ± 0.01** | 105.8 ± 0.7 | 2.36 ± 0.01 | 73.9 ± 0.8 | 1.65 ± 0.01 | **0.91 ± 0.01** | 4.78 ± 0.01 |
| | | | Cache (AB) | **0.82 ± 0.00** | **102.6 ± 0.5** | **2.30 ± 0.02** | **71.2 ± 0.4** | **1.60 ± 0.01** | **0.91 ± 0.00** | **4.79 ± 0.01** |
| | Tabasco (hot) | - | Base 18 | 0.86 ± 0.00 | 92.1 ± 0.3 | 3.59 ± 0.01 | 54.6 ± 0.2 | 2.08 ± 0.01 | 0.94 ± 0.01 | 4.89 ± 0.00 |
| | | 3 | Cache (TS) | 0.88 ± 0.00 | 81.5 ± 0.4 | 3.23 ± 0.02 | **46.7 ± 0.4** | **1.80 ± 0.01** | **0.84 ± 0.00** | 4.90 ± 0.00 |
| | | | Cache (AB) | **0.89 ± 0.00** | **80.6 ± 0.2** | **3.20 ± 0.01** | 46.8 ± 0.4 | 1.81 ± 0.01 | 0.85 ± 0.00 | **4.91 ± 0.01** |
| | | - | Base 14 | 0.00 ± 0.00 | x | x | x | x | x | x |
| | | 4 | Cache (TS) | **0.88 ± 0.00** | **93.1 ± 0.8** | **3.66 ± 0.02** | **59.2 ± 0.6** | **2.27 ± 0.02** | **0.87 ± 0.00** | 4.81 ± 0.01 |
| | | | Cache (AB) | **0.88 ± 0.00** | 95.7 ± 0.5 | 3.77 ± 0.02 | 63.0 ± 0.3 | 2.42 ± 0.01 | **0.87 ± 0.00** | **4.83 ± 0.01** |

| | | D | Mode | Vina ↓ | VinaMin ↓ | Strain ↓ | PB Validity ↑ | Angle W1 ↓ | Length W1 ↓ | Dihedral W1 ↓ |
|---|---|---|---|---|---|---|---|---|---|---|
| SPINDR | FLOWR | - | Base 34 | -6.82 ± 0.94 | -7.13 ± 0.93 | 109.97 ± 62.22 | 0.91 ± 0.25 | **1.290** | 0.00744 | 4.325 |
| | | 3 | Cache (TS) | **-6.93 ± 0.93** | **-7.22 ± 0.92** | 104.47 ± 65.44 | **0.93 ± 0.23** | 1.297 | **0.00572** | 3.990 |
| | | | Cache (AB) | -6.92 ± 0.93 | **-7.22 ± 0.91** | **98.35 ± 57.73** | **0.93 ± 0.22** | 1.297 | **0.00572** | **3.938** |
| | | - | Base 26 | -6.77 ± 0.93 | -7.09 ± 0.93 | 119.36 ± 68.33 | 0.90 ± 0.27 | 1.388 | 0.00650 | 4.502 |
| | | 4 | Cache (TS) | **-6.89 ± 0.92** | **-7.18 ± 0.92** | 111.96 ± 67.55 | 0.91 ± 0.25 | 1.403 | **0.00581** | 4.278 |
| | | | Cache (AB) | -6.87 ± 0.92 | -7.16 ± 0.92 | **103.16 ± 61.31** | **0.92 ± 0.23** | **1.217** | 0.00627 | **4.201** |
| | FLOWR.root | - | Base 34 | -7.32 ± 0.82 | -7.55 ± 0.85 | 53.75 ± 37.22 | 0.96 ± 0.16 | 0.440 | 0.00867 | 4.028 |
| | | 3 | Cache (TS) | -7.34 ± 0.82 | -7.54 ± 0.82 | **49.86 ± 35.68** | **0.98 ± 0.09** | 0.294 | 0.00852 | **3.451** |
| | | | Cache (AB) | **-7.36 ± 0.82** | **-7.57 ± 0.83** | 50.14 ± 35.62 | **0.98 ± 0.09** | **0.293** | 0.00842 | 3.641 |
| | | - | Base 26 | -7.29 ± 0.84 | -7.53 ± 0.85 | 58.90 ± 40.81 | 0.94 ± 0.20 | 0.568 | 0.00906 | 4.451 |
| | | 4 | Cache (TS) | **-7.32 ± 0.81** | -7.53 ± 0.81 | **51.70 ± 36.61** | **0.98 ± 0.11** | 0.306 | **0.00848** | **3.634** |
| | | | Cache (AB) | **-7.32 ± 0.81** | **-7.54 ± 0.83** | 52.57 ± 37.22 | **0.98 ± 0.12** | 0.328 | 0.00848 | 3.845 |
| Crossdocked | FLOWR.root | - | Base 34 | -7.36 ± 0.53 | -7.90 ± 0.43 | 45.71 ± 33.16 | 0.89 ± 0.25 | **1.326** | 0.00422 | 4.055 |
| | | 3 | Cache (TS) | -7.40 ± 0.54 | -7.92 ± 0.43 | **42.45 ± 31.45** | **0.91 ± 0.21** | 1.447 | 0.00482 | **3.808** |
| | | | Cache (AB) | **-7.41 ± 0.54** | **-7.94 ± 0.43** | 42.60 ± 31.59 | **0.91 ± 0.21** | 1.443 | **0.00414** | 3.776 |
| | | - | Base 26 | -7.32 ± 0.53 | -7.89 ± 0.43 | 51.12 ± 39.43 | 0.86 ± 0.29 | **1.343** | **0.00364** | 4.331 |
| | | 4 | Cache (TS) | **-7.36 ± 0.54** | **-7.90 ± 0.43** | **44.20 ± 32.91** | **0.90 ± 0.22** | 1.423 | 0.00552 | **3.850** |
| | | | Cache (AB) | **-7.36 ± 0.54** | **-7.90 ± 0.43** | 44.95 ± 33.79 | 0.89 ± 0.23 | 1.390 | 0.00433 | 4.102 |

## 5.1 Caching allows for faster inference at little to no quality loss

We compare base models with default inference configurations (Base 100 for SemlaFlow, FLOWR and FLOWR.root, Base 50 for Tabasco) against cached variants using a caching interval of $D = 2$. Additionally, we include non-cached baselines using uniform step reduction (Base 51 for SemlaFlow, FLOWR and FLOWR.root, Base 26 for Tabasco). These reduced budgets are chosen to align with the effective number of function evaluations performed by the cached variants. We report key performance indicators in Table 1 and show additional generation quality metrics in Appendix D and Appendix E. Additionally, we discuss the impact of the caching order as well as the number of inference steps on the generation quality in Appendix B.

The evaluated caching strategies enable a 2× increase in throughput while maintaining, and in several cases exceeding, the original sample quality. For instance, in the SemlaFlow experiments, both Taylor (TS) and Adams–Bashforth (AB) caching at 51 steps achieve lower Energy and Strain than the 100-step model. In contrast, reducing steps without caching leads to immediate performance degradation.

A similar trend appears in the FLOWR and FLOWR.root results across both evaluated datasets. Caching at about half the step budget achieves the same Vina Score and Posebusters Validity as the 100-step model. However, uniform step reduction results in a drop in both metrics. These results show that predictive feature caching is an effective approach for generating high-quality samples at lower computational cost.

## 5.2 Caching enables high-fidelity generation under strict computational constraints

The performance gap between feature caching and uniform step reduction widens as computational constraints become more stringent. Although quality degradation eventually occurs at extreme operating points of 1/3 or 1/4 of the original inference budget, caching demonstrates greater robustness compared to the non-cached baseline. As indicated in Table 2, the Tabasco model reaches a breakdown point at 14 steps,

failing to produce any valid molecules. In contrast, both Taylor (TS) and Adams–Bashforth (AB) caching variants maintain high validity while preserving physically meaningful energy and strain metrics.

For SemlaFlow, although the base model remains valid at 26 steps, its Energy and Strain increase, whereas caching at the same budget recovers significantly better geometries.

In the pocket-conditioned generation experiments, caching at $D = 4$ (26 steps) nearly matches the Vina Score and Pose-busters Validity of the more computationally expensive 100-step base model, whereas the uniform reduction baseline clearly declines in sampling quality. The evaluation across multiple architectures (FLOWR.root, FLOWR) and diverse datasets (SPINDR, CrossDocked2020) demonstrates that caching consistently outperforms step-reduction baselines, providing a robust mechanism for inference acceleration.

### 5.3 Caching
**improves sampling quality at the base model's budget**

While caching is primarily utilized for acceleration, our results demonstrate that it can also enhance sample quality beyond the performance of the base model at equivalent computational budgets. As shown in Figure 3, a cached variant with an effective budget of 100 steps (derived from 200 total steps with a

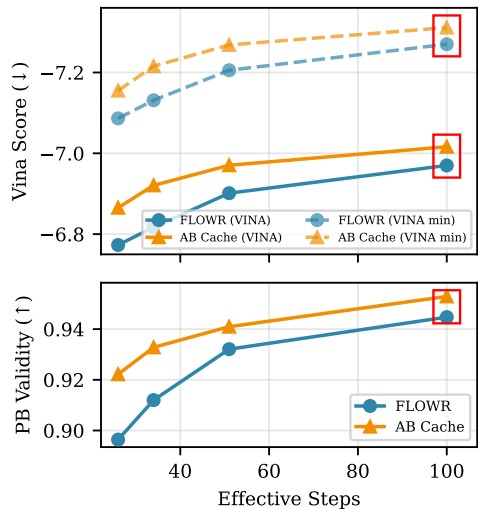

Figure 3: Comparison of VINA Score and Validity across inference budgets.

caching interval of $D = 2$) consistently outperforms the standard 100-step base model across all key metrics. The cached model achieves superior Vina Score, minimized Vina Score and Posebusters Validity compared to the base model at the same effective step count. These results indicate that leveraging historical feature states, such as through Adams–Bashforth forecasting, allows the model to maintain higher trajectory fidelity, yielding quality gains that are inaccessible to the base model through standard sampling alone.

### 5.4 Caching is compatible with general inference acceleration methods

Predictive feature caching, as presented in this work, is complementary to lossless inference-time optimizations. We pair AB caching with graph compilation of the SemlaFlow backbone $v_\theta$ (Paszke et al., 2019). The runtime branch that selects between evaluating $F(h_t)$ and using an approximation $\widehat{F}(h_{t_{k+r}})$ introduces control flow that would break whole-graph compilation. However, this can be easily avoided by compiling the backbone $v_\theta$ and keeping the selection logic outside the compiled region. We also combine this with TensorFloat-32 (TF32) matrix-multiply kernels instead of standard FP32 computation to further increase throughput without significantly affecting evaluation metrics.

In Figure 4, we report inference time and peak memory. Caching incurs a modest increase in peak memory due to maintaining $C(h_t)$, whereas compilation lowers peak memory slightly. Caching alone yields $\sim$ 3x faster inference; combined with compilation and TF32, the speedup reaches up to 7x. This reduces the time to generate 10,000 molecules from >14 min to $\sim$2 min, with no significant loss in sampling quality.

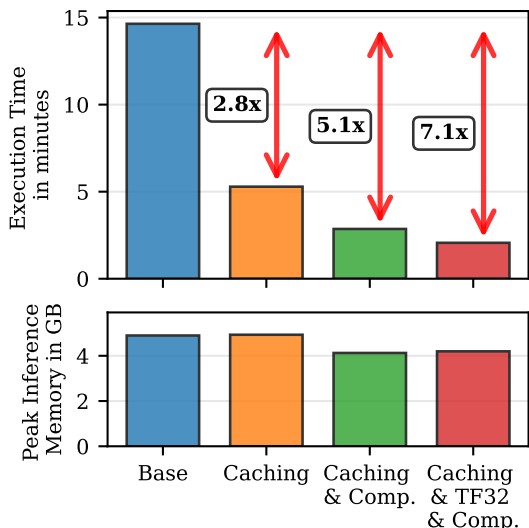

Figure 4: SemlaFlow inference time and memory overhead to sample 10,000 molecular geometries of various acceleration method combinations.

## 6 Conclusion

In this paper, we address inference latency in molecular geometry generation by adapting a training-free predictive caching scheme to forecast intermediate hidden states during sampling for the molecular domain. Empirical evaluations quantify the speed–quality trade-off and demonstrate up to threefold reductions in wall-clock inference time while maintaining comparable conformer quality. More broadly, this work seeks to motivate systematic discussion of inference-time efficiency for molecule generation and to identify strategies for scaling to millions of samples.

## Acknowledgements

We are grateful to Simon Langrieger, Gaspar Rochette, and Lennard Schaub for valuable feedback and discussions, which have greatly improved the quality of this paper, as well as to Begüm Çığ for her guidance on efficiency metrics.

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

# A    Inference Hyperparameters

Inference with the SemlaFlow model was performed using a batch cost budget of 8192, rather than a fixed number of molecules per batch, consistent with the approach described by Irwin et al. (2025). In this configuration, the batch cost is allocated proportionally to molecule size, which maintains stable GPU memory usage and consistent compute time across batches containing molecules of varying sizes. The continuous dynamics of the base model are solved with 100 inference steps, and ODE sampling uses a logarithmic time discretization. For categorical variables, Irwin et al. (2025) inject sampling noise with a noise level of 1.0, which controls the stochasticity of categorical sampling during inference.

All inference results for the Tabasco base model use 50 inference steps (Vonessen et al., 2025). While the base configuration of the Tabasco model in Vonessen et al. (2025) is presented with 100 inference steps, ablations in this work demonstrate that "additional steps have no effect on molecular quality"; measured by the Validity, Novelty and Connectivity of the generated molecules. Following this insight, we adopt the 50-step configuration as our primary baseline to ensure a more rigorous evaluation. By targeting the model's true efficiency frontier rather than a redundant default budget, we demonstrate that predictive caching provides genuine acceleration. A maximum batch size of 100 samples is used during inference. We report results for the "hot" model variant (Vonessen et al., 2025), corresponding to the medium-sized configuration with approximately 15M parameters.

For the structure-based molecule generation models FLOWR (Cremer et al., 2025) and FLOWR.root (Cremer et al., 2026), results are obtained using 100 inference steps with a linear time discretization schedule. Protein pocket coordinates are kept fixed during generation, and hydrogen atoms are removed from protein and ligand structures for both the SPINDR and the CrossDocked2020 dataset which models heavy atoms only. Coordinate trajectories are deterministic, with no Gaussian noise added to spatial coordinates, and no corrector steps are applied, resulting in an Euler sampling scheme. Discrete atom and bond types are sampled using a uniform-sample categorical strategy with a categorical sampling noise level of 1. Dynamic batching is controlled via a batch cost budget of 100, with molecule cost scaling quadratically with the number of atoms to reflect attention-based memory usage.

Table 3: Optimal caching order by model, caching method and interval.

|  | Interval | SemlaFlow | Tabasco | FLOWR | FLOWR.root |
|---|---|---|---|---|---|
| TaylorSeer | $D = 2$ | 1 | 3 | 1 | 2 |
|  | $D = 3$ | 1 | 2 | 1 | 2 |
|  | $D = 4$ | 1 | 3 | 1 | 2 |
| Adams–Bashforth | $D = 2$ | 3 | 3 | 2 | 3 |
|  | $D = 3$ | 3 | 3 | 2 | 3 |
|  | $D = 4$ | 3 | 2 | 2 | 3 |

We report the optimal caching orders for TaylorSeer ($m$) and Adams–Bashforth ($j$) in Table 3, determined empirically via hyperparameter optimization across the evaluated models and caching intervals. To determine the optimal caching orders, we conduct a systematic grid search on reduced sample sets. For unconditional generation, each hyperparameter configuration was evaluated by sampling 1,000 molecules. In structure-based generation, the optimization sweep used a subset of 20 protein pockets, sampling 100 ligands per pocket to identify the most effective settings. The search space for these experiments included caching orders $m \in \{1, 2, 3\}$ for TaylorSeer and $j \in \{2, 3, 4\}$ for the Adams-Bashforth protocol across all evaluated intervals $D$. Empirical observations indicate that although the optimal order is sensitive to the specific model architecture and caching mode, it typically remains consistent across different caching intervals. Final configurations were selected based on core metrics relevant to each scenario. Hyperparameters for unconditional models were chosen based on their Posebusters Validity, Strain, and Energy to ensure physical plausibility. For structure-based generation, a combination of Vina Score, Minimized Vina, Strain, and Posebusters Validity was used to balance binding affinity with geometric realism. These criteria provide a robust objective for the hyperparameter optimization, and they may be further weighted or adjusted depending on the specific priorities of the molecular generation scenario.

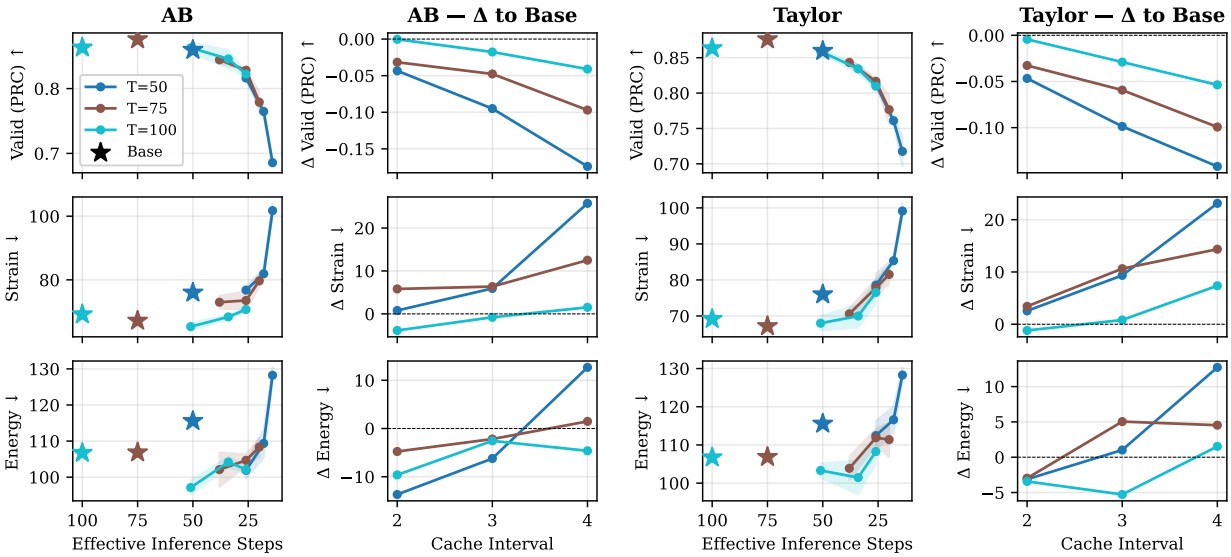

Figure 5: Ablation of the number of steps and caching intervals on SemlaFlow. Performance metrics are reported across 1,000 samples (3 random seeds) for varying base steps $T$ and caching intervals $D$.

# B Ablation Studies

## B.1 Effect of the Initial Number of Steps

To investigate the interplay between the number of steps (i.e., the discretization resolution) and caching frequency, we conduct an ablation study using the SemlaFlow model in Figure 5. Our results indicate that for a fixed computational budget defined by the number of effective inference steps, higher base temporal resolutions ($T$) paired with larger caching intervals ($D$) generally yield superior performance; for instance, the ($T = 100, D = 4$) configuration outperforms ($T = 50, D = 2$). Furthermore, sensitivity analysis relative to non-cached baselines reveals that models with $T = 100$ exhibit the greatest robustness to quality degradation.

We note, however, a marginal trade-off involving "cache staleness": at identical effective budgets, configurations with shorter intervals (e.g., $T = 75, D = 3$) can occasionally surpass those with longer intervals (e.g., $T = 100, D = 4$).

## B.2 Effect of the Caching Order

Additionally, we evaluate the sensitivity of our approach to the forecasting order in Figure 6. For AB caching, our results reveal a non-monotonic relationship between order and performance: while $j = 3$ consistently improves validity and minimizes energy/strain relative to $j = 2$, higher-order approximations ($j = 4$) lead to unsatisfactory results (validity approximately 50%). Taylor forecasting shows a consistent performance ranking of orders across the evaluated metrics, albeit sensitive to the specific caching interval. These findings suggest that predictive caching is generally robust with respect to the caching order, but the forecasting order can serve as a hyperparameter for finetuning the sampling quality.

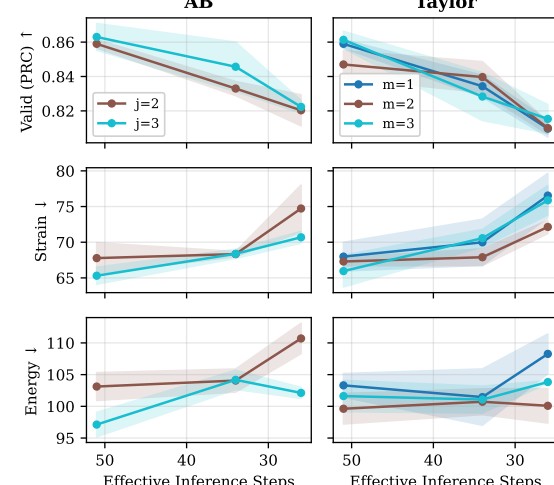

Figure 6: Impact of forecasting order $j$ on caching performance.

Table 4: Comparison of uniform step reduction as well as inference with predictive feature caching in combination with the Euler and Heun solver.

| D | Mode | NFE | Valid (PRC) ↑ | Energy ↓ | Energy p.A ↓ | Strain ↓ | Strain p.A ↓ | Opt. RMSD ↓ | Lipinski ↑ |
|---|------|-----|---------------|----------|--------------|----------|--------------|-------------|------------|
| - | Base 100 (Euler) | 100 | 0.88 ± 0.01 | 108.8 ± 0.9 | 2.38 ± 0.01 | 69.6 ± 0.7 | 1.50 ± 0.01 | 0.86 ± 0.00 | **4.82 ± 0.00** |
| - | Base 50 (Heun) | 100 | **0.90 ± 0.00** | **107.9 ± 1.1** | **2.34 ± 0.02** | **68.2 ± 0.9** | **1.46 ± 0.01** | **0.83 ± 0.00** | **4.82 ± 0.00** |
| - | Base 51 (Euler) | 51 | 0.86 ± 0.01 | 115.5 ± 0.8 | 2.51 ± 0.01 | 75.9 ± 0.8 | 1.63 ± 0.01 | 0.88 ± 0.00 | **4.82 ± 0.00** |
| - | Base 26 (Heun) | 52 | **0.88 ± 0.00** | 112.5 ± 0.1 | 2.44 ± 0.00 | 74.9 ± 0.4 | 1.60 ± 0.00 | 0.85 ± 0.00 | 4.80 ± 0.00 |
| 2 | Cache (TS) + Euler | 51 | 0.85 ± 0.00 | 103.1 ± 1.2 | 2.28 ± 0.02 | 67.5 ± 1.0 | 1.48 ± 0.01 | 0.87 ± 0.00 | **4.82 ± 0.00** |
|   | Cache (AB) + Euler | 51 | 0.87 ± 0.00 | **96.5 ± 0.9** | **2.15 ± 0.01** | **62.8 ± 0.6** | **1.40 ± 0.01** | 0.87 ± 0.01 | 4.79 ± 0.00 |
|   | Cache (AB) + Heun | 51 | **0.88 ± 0.01** | 99.5 ± 0.6 | 2.21 ± 0.01 | 64.4 ± 0.9 | 1.42 ± 0.02 | **0.84 ± 0.01** | 4.81 ± 0.00 |
| - | Base 34 (Euler) | 34 | 0.85 ± 0.00 | 120.3 ± 1.6 | 2.62 ± 0.03 | 82.0 ± 1.1 | 1.78 ± 0.02 | 0.90 ± 0.01 | 4.80 ± 0.00 |
| - | Base 17 (Heun) | 34 | **0.86 ± 0.00** | 110.8 ± 0.6 | 2.44 ± 0.02 | 74.2 ± 0.2 | 1.64 ± 0.00 | 0.89 ± 0.00 | 4.80 ± 0.00 |
| 3 | Cache (TS) + Euler | 34 | 0.85 ± 0.01 | 103.8 ± 0.5 | 2.31 ± 0.01 | 70.0 ± 0.5 | 1.56 ± 0.01 | 0.89 ± 0.00 | 4.78 ± 0.00 |
|   | Cache (AB) + Euler | 34 | 0.85 ± 0.00 | **100.5 ± 1.0** | **2.25 ± 0.02** | 67.7 ± 0.7 | 1.51 ± 0.02 | 0.90 ± 0.01 | 4.80 ± 0.00 |
|   | Cache (AB) + Heun | 34 | **0.86 ± 0.00** | 102.8 ± 1.1 | 2.28 ± 0.02 | **66.9 ± 0.9** | **1.49 ± 0.02** | **0.87 ± 0.00** | **4.82 ± 0.00** |
| - | Base 26 (Euler) | 26 | 0.82 ± 0.00 | 123.5 ± 0.4 | 2.69 ± 0.01 | 85.5 ± 0.5 | 1.85 ± 0.01 | 0.92 ± 0.00 | **4.79 ± 0.00** |
| - | Base 13 (Heun) | 26 | **0.84 ± 0.00** | 112.1 ± 0.4 | 2.47 ± 0.01 | 77.2 ± 0.4 | 1.71 ± 0.01 | 0.90 ± 0.00 | 4.78 ± 0.00 |
| 4 | Cache (TS) + Euler | 26 | 0.76 ± 0.00 | 133.0 ± 1.1 | 2.93 ± 0.02 | 98.2 ± 0.8 | 2.17 ± 0.01 | 0.90 ± 0.00 | 4.69 ± 0.01 |
|   | Cache (AB) + Euler | 26 | 0.82 ± 0.00 | **102.6 ± 0.5** | **2.30 ± 0.02** | **71.2 ± 0.4** | **1.60 ± 0.01** | 0.91 ± 0.00 | **4.79 ± 0.01** |
|   | Cache (AB) + Heun | 26 | **0.83 ± 0.00** | 109.5 ± 0.7 | 2.43 ± 0.01 | 75.2 ± 0.8 | 1.67 ± 0.02 | **0.88 ± 0.00** | 4.78 ± 0.00 |

## B.3 Comparison against Heun Solver

We compare predictive feature caching against the Heun solver, a second-order Runge-Kutta method known for its optimal balance between truncation error and sampling efficiency (Karras et al., 2022). In contrast to first-order Euler methods, Heun uses a predictor-corrector scheme. It computes an initial Euler step and refines it using the average gradient between noise levels. This increased precision requires two neural network forward passes per diffusion step, doubling the computational cost per iteration.

As shown in Table 4, the Heun solver applied to the SemlaFlow model consistently outperforms the Euler solver across all quality metrics when computational resources are unconstrained. However, applying predictive feature caching at inference time maintains higher sampling quality than Heun when the number of function evaluations (NFEs) is reduced.

Integrating Heun with AB caching achieves the best performance balance across all metrics, demonstrating that caching can complement advanced solvers and can be used in combination to achieve state-of-the-art Pareto efficiency.

## B.4 Layer-Selective Caching

We conduct an ablation study on layer-selective caching using the SemlaFlow model. While our primary approach employs a last-block forecast to bypass the entire network prefix, Table 5 presents a hybrid method in which only the first half of the $L$ blocks are cached (denoted with *). For the initial layers $F^1$ through $F^{L/2}$, caching is applied with interval $D$, whereas the remaining layers $F^{L/2+1}$ through $F^L$ are fully evaluated at every step $t$. This "first-half" caching strategy is compared against the standard last-block forecast and the uniform step-reduction baseline.

The results in Table 5 indicate that, across all evaluated step budgets, layer-selective caching consistently underperforms relative to the last-block caching strategy. This performance gap may be attributed to the increased difficulty and noise susceptibility of predicting intermediate representations within the backbone, compared to the terminal features of the last block. While the final outputs of the network evolve along a highly regularized path to produce the velocity field, the internal hidden states can exhibit higher-frequency variations that can not be captured well by predictive feature caching.

Table 5: Ablation of layer-selective caching on SemlaFlow. Last-block forecasting is compared against caching only the first $L/2$ layers (denoted by *).

| $D$ | Mode | Valid (PRC) ↑ | Energy ↓ | Energy p.A ↓ | Strain ↓ | Strain p.A ↓ | Opt. RMSD ↓ | Lipinski ↑ | Throughput ↑ |
|---|---|---|---|---|---|---|---|---|---|
| - | Base 100 | $0.88 \pm 0.01$ | $108.8 \pm 0.9$ | $2.38 \pm 0.01$ | $69.6 \pm 0.7$ | $1.50 \pm 0.01$ | $0.86 \pm 0.00$ | $4.82 \pm 0.00$ | $11.4 \pm 0.1$ |
| - | Base 51 | $0.86 \pm 0.01$ | $115.5 \pm 0.8$ | $2.51 \pm 0.01$ | $75.9 \pm 0.8$ | $1.63 \pm 0.01$ | $0.88 \pm 0.00$ | $4.82 \pm 0.00$ | $21.9 \pm 0.2$ |
| 2 | Cache (TS) | $0.85 \pm 0.00$ | $103.1 \pm 1.2$ | $2.28 \pm 0.02$ | $67.5 \pm 1.0$ | $1.48 \pm 0.01$ | $0.87 \pm 0.00$ | $4.82 \pm 0.00$ | $21.8 \pm 0.2$ |
| | Cache (TS)* | $0.84 \pm 0.00$ | $115.5 \pm 0.5$ | $2.53 \pm 0.01$ | $76.7 \pm 0.4$ | $2.53 \pm 0.01$ | $0.86 \pm 0.00$ | $4.80 \pm 0.00$ | $15.1 \pm 0.4$ |
| | Cache (AB) | $0.87 \pm 0.00$ | $96.5 \pm 0.9$ | $2.15 \pm 0.01$ | $62.8 \pm 0.6$ | $1.40 \pm 0.01$ | $0.87 \pm 0.01$ | $4.79 \pm 0.00$ | $22.1 \pm 0.0$ |
| | Cache (AB)* | $0.83 \pm 0.00$ | $130.2 \pm 1.0$ | $2.85 \pm 0.02$ | $94.8 \pm 0.7$ | $2.07 \pm 0.01$ | $0.79 \pm 0.01$ | $4.57 \pm 0.01$ | $15.1 \pm 0.0$ |
| - | Base 34 | $0.85 \pm 0.00$ | $120.3 \pm 1.6$ | $2.62 \pm 0.03$ | $82.0 \pm 1.1$ | $1.78 \pm 0.02$ | $0.90 \pm 0.01$ | $4.80 \pm 0.00$ | $33.1 \pm 0.2$ |
| 3 | Cache (TS) | $0.85 \pm 0.01$ | $103.8 \pm 0.5$ | $2.31 \pm 0.01$ | $70.0 \pm 0.5$ | $1.56 \pm 0.01$ | $0.89 \pm 0.00$ | $4.78 \pm 0.00$ | $32.4 \pm 0.3$ |
| | Cache (TS)* | $0.80 \pm 0.00$ | $122.3 \pm 0.9$ | $2.69 \pm 0.02$ | $85.4 \pm 1.2$ | $1.88 \pm 0.03$ | $0.88 \pm 0.00$ | $4.74 \pm 0.00$ | $16.8 \pm 0.4$ |
| | Cache (AB) | $0.85 \pm 0.00$ | $100.5 \pm 1.0$ | $2.25 \pm 0.02$ | $67.7 \pm 0.7$ | $1.51 \pm 0.02$ | $0.90 \pm 0.01$ | $4.80 \pm 0.00$ | $32.1 \pm 0.2$ |
| | Cache (AB)* | $0.79 \pm 0.01$ | $146.8 \pm 1.5$ | $3.24 \pm 0.04$ | $111.1 \pm 0.8$ | $2.47 \pm 0.02$ | $0.81 \pm 0.01$ | $4.49 \pm 0.00$ | $16.8 \pm 0.2$ |
| - | Base 26 | $0.82 \pm 0.00$ | $123.5 \pm 0.4$ | $2.69 \pm 0.01$ | $85.5 \pm 0.5$ | $1.85 \pm 0.01$ | $0.92 \pm 0.00$ | $4.79 \pm 0.00$ | $42.2 \pm 0.4$ |
| 4 | Cache (TS) | $0.76 \pm 0.00$ | $133.0 \pm 1.1$ | $2.93 \pm 0.02$ | $98.2 \pm 0.8$ | $2.17 \pm 0.01$ | $0.90 \pm 0.00$ | $4.69 \pm 0.01$ | $41.5 \pm 0.3$ |
| | Cache (TS)* | $0.82 \pm 0.01$ | $105.8 \pm 0.7$ | $2.36 \pm 0.01$ | $73.9 \pm 0.8$ | $1.65 \pm 0.01$ | $0.91 \pm 0.01$ | $4.78 \pm 0.00$ | $17.7 \pm 0.3$ |
| | Cache (AB) | $0.82 \pm 0.00$ | $102.6 \pm 0.5$ | $2.30 \pm 0.02$ | $71.2 \pm 0.4$ | $1.60 \pm 0.01$ | $0.91 \pm 0.00$ | $4.79 \pm 0.01$ | $41.2 \pm 0.4$ |
| | Cache (AB)* | $0.74 \pm 0.00$ | $160.5 \pm 0.6$ | $3.57 \pm 0.02$ | $126.8 \pm 0.3$ | $2.83 \pm 0.01$ | $0.84 \pm 0.00$ | $4.41 \pm 0.00$ | $17.7 \pm 0.2$ |

## B.5 Caching Forecasting Error

To evaluate the precision of the different caching strategies, we measure the forecasting error across the generation trajectory for the SemlaFlow model. This error is quantified using the Root Mean Square Error (RMSE) between the predicted feature $\hat{F}(h_t)$ (in this case, the coordinates) and the ground-truth backbone evaluation $F(h_t)$. Figure 7 presents the error for both Taylor and Adams-Bashforth (AB) caching across varying caching intervals $D \in \{2, 3, 4\}$.

The results show the error accumulating between cache refreshes and returning to zero at each backbone evaluation. For all caching strategies, the error magnitude increases with the caching interval $D$, reflecting the challenge of long-range forecasting as the feature trajectory diverges from the local expansion. The forecasting error is not uniform across the time $t$; error peaks occur during the intermediate stages of the generation process. The error diminishes significantly as $t \to 1.0$. This stabilization indicates that as the model converges toward a final molecular conformer, the feature trajectory becomes increasingly predictable.

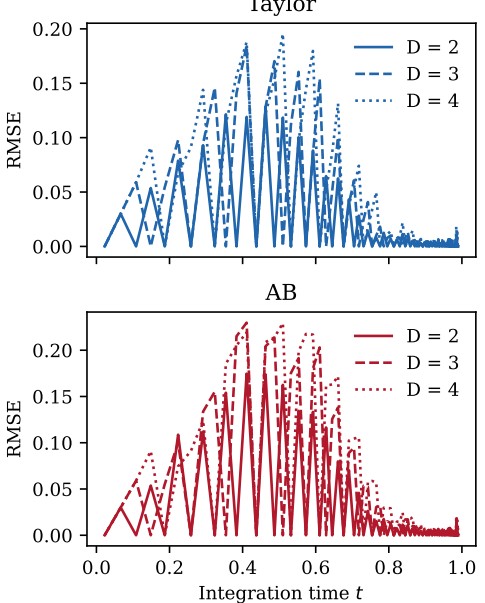

Figure 7: Temporal evolution of forecasting error.

## C Generation Quality Metrics

We evaluate sample quality using standard graph-level metrics: Novelty (fraction of generated molecules not seen in the training set), and Uniqueness (fraction of distinct molecules under canonical SMILES). As these topology-only metrics are saturated by current models and do not indicate conformation quality, we follow Irwin et al. (2025) and report (per-atom) energy and (per-atom) strain in kcal∗mol$^{-1}$, where lower strain or energy indicates higher plausibility. In a similar vein, we compute the root-mean-square deviation between the generated conformer and its energy-optimized counterpart. To assess physical validity beyond topology we follow the evaluation from Buttenschoen et al. (2025), also used in Vonessen et al. (2025), and report validity (Posebusters, RDKit, Connected = PRC) as the fraction of molecules that are connected, can be sanitized by RDKit (Landrum, 2013), and pass all Posebusters checks. Posebusters (Buttenschoen et al.,

Table 6: Comparison of the SemlaFlow and Tabasco base models and their cached variants with additional quality metrics.

| | $D$ | Mode | LogP | Novelty | QED | Uniqueness | Validity | Conn. Validity |
|---|---|---|---|---|---|---|---|---|
| Semla Flow | - | Base 100 | $2.69 \pm 0.01$ | $1.00 \pm 0.00$ | $0.68 \pm 0.00$ | $1.00 \pm 0.00$ | $0.94 \pm 0.00$ | $0.92 \pm 0.00$ |
| | - | Base 51 | $2.54 \pm 0.01$ | $1.00 \pm 0.00$ | $0.67 \pm 0.00$ | $1.00 \pm 0.00$ | $0.94 \pm 0.00$ | $0.90 \pm 0.00$ |
| | 2 | Cache (TS) | $2.54 \pm 0.01$ | $1.00 \pm 0.00$ | $0.66 \pm 0.00$ | $0.99 \pm 0.00$ | $0.94 \pm 0.00$ | $0.91 \pm 0.00$ |
| | | Cache (AB) | $2.50 \pm 0.02$ | $1.00 \pm 0.00$ | $0.64 \pm 0.00$ | $1.00 \pm 0.00$ | $0.94 \pm 0.00$ | $0.92 \pm 0.00$ |
| | - | Base 34 | $2.32 \pm 0.02$ | $1.00 \pm 0.00$ | $0.66 \pm 0.00$ | $1.00 \pm 0.00$ | $0.94 \pm 0.00$ | $0.89 \pm 0.00$ |
| | 3 | Cache (TS) | $2.38 \pm 0.02$ | $1.00 \pm 0.00$ | $0.64 \pm 0.00$ | $1.00 \pm 0.00$ | $0.93 \pm 0.00$ | $0.89 \pm 0.01$ |
| | | Cache (AB) | $2.39 \pm 0.01$ | $1.00 \pm 0.00$ | $0.64 \pm 0.00$ | $1.00 \pm 0.00$ | $0.93 \pm 0.00$ | $0.90 \pm 0.00$ |
| | - | Base 26 | $2.17 \pm 0.00$ | $1.00 \pm 0.00$ | $0.65 \pm 0.00$ | $1.00 \pm 0.00$ | $0.94 \pm 0.00$ | $0.88 \pm 0.00$ |
| | 4 | Cache (TS) | $2.24 \pm 0.02$ | $1.00 \pm 0.00$ | $0.63 \pm 0.00$ | $1.00 \pm 0.00$ | $0.92 \pm 0.01$ | $0.87 \pm 0.01$ |
| | | Cache (AB) | $2.26 \pm 0.01$ | $1.00 \pm 0.00$ | $0.63 \pm 0.00$ | $1.00 \pm 0.00$ | $0.92 \pm 0.00$ | $0.88 \pm 0.00$ |
| Tabasco (hot) | - | Base 50 | $2.87 \pm 0.01$ | $1.00 \pm 0.00$ | $0.64 \pm 0.00$ | $0.99 \pm 0.00$ | $0.97 \pm 0.00$ | $0.97 \pm 0.00$ |
| | - | Base 26 | $2.83 \pm 0.01$ | $1.00 \pm 0.00$ | $0.64 \pm 0.00$ | $1.00 \pm 0.00$ | $0.96 \pm 0.00$ | $0.96 \pm 0.00$ |
| | 2 | Cache (TS) | $2.95 \pm 0.02$ | $1.00 \pm 0.00$ | $0.64 \pm 0.00$ | $0.99 \pm 0.00$ | $0.97 \pm 0.00$ | $0.97 \pm 0.00$ |
| | | Cache (AB) | $2.91 \pm 0.02$ | $1.00 \pm 0.00$ | $0.64 \pm 0.00$ | $0.99 \pm 0.00$ | $0.97 \pm 0.00$ | $0.97 \pm 0.00$ |
| | - | Base 18 | $2.94 \pm 0.01$ | $1.00 \pm 0.00$ | $0.65 \pm 0.00$ | $1.00 \pm 0.00$ | $0.95 \pm 0.00$ | $0.95 \pm 0.00$ |
| | 3 | Cache (TS) | $2.97 \pm 0.02$ | $1.00 \pm 0.00$ | $0.63 \pm 0.00$ | $1.00 \pm 0.00$ | $0.97 \pm 0.00$ | $0.97 \pm 0.00$ |
| | | Cache (AB) | $2.89 \pm 0.02$ | $1.00 \pm 0.00$ | $0.63 \pm 0.00$ | $1.00 \pm 0.00$ | $0.97 \pm 0.00$ | $0.97 \pm 0.00$ |
| | - | Base 14 | x | x | x | x | x | x |
| | 4 | Cache (TS) | $3.11 \pm 0.01$ | $1.00 \pm 0.00$ | $0.60 \pm 0.00$ | $1.00 \pm 0.00$ | $0.97 \pm 0.00$ | $0.96 \pm 0.00$ |
| | | Cache (AB) | $3.02 \pm 0.01$ | $1.00 \pm 0.00$ | $0.60 \pm 0.00$ | $1.00 \pm 0.00$ | $0.97 \pm 0.00$ | $0.97 \pm 0.00$ |

2024) requires passing checks on bond lengths/angles, planarity (aromatics and double bonds), steric clashes, and internal energy. In terms of properties of the generated molecules, we evaluate QED quantifying how "drug-like" a molecule is based on several physicochemical properties (Bickerton et al., 2012), the Lipinski score representing the average number of Lipinski's "Rule of Five" criteria satisfied by the generated samples (Lipinski et al., 1997), and the octanol-water partition coefficient LogP (Wildman and Crippen, 1999).

In addition to Validity, Novelty, Strain, QED, the Lipinski score and LogP as detailed above, we closely follow Cremer et al. (2025) in evaluating the quality of the samples generated by the FLOWR model by using a comprehensive suite of metrics encompassing binding affinity, geometric realism, and chemical diversity. Binding performance is measured using the Vina Score and its locally minimized counterpart, Minimized Vina, which estimate ligand binding energy (Trott and Olson, 2010; Baillif et al., 2024). Wasserstein-1 (W1) distances for bond lengths, angles, and dihedrals quantify the alignment between generated distributions and the SPINDR dataset's local and torsional geometry. The model's exploratory capacity is assessed through uniqueness and diversity metrics in both 2D graph and 3D conformer spaces.

Additionally, the generated distribution is examined for physicochemical properties by reporting synthetic accessibility (SA), molecular weight (MolWt), LogP, and counts of specific chemical features, including rings, aromatic rings, hydrogen bond acceptors (HAcceptors), and donors (HDonors).

## D  Additional Results: Unconditional Generation

Table 6 presents additional evaluation metrics for the unconditional SemlaFlow and Tabasco models, focusing on distributional properties and chemical descriptors. In terms of Novelty and Uniqueness, we observe no meaningful differences between the base models and their cached counterparts, with all variants maintaining scores near 1.00. Regarding QED, the base models show slightly better scores as the number of inference steps is reduced. The cached variants of SemlaFlow maintain Connectivity and Validity at levels comparable to

Table 7: Comparison of the FLOWR and FLOWR.root models and their cached variants with additional diversity metrics.

| | $D$ | Mode | Validity | Novelty | Uniq. 2D | Uniq. 3D | Diversity 2D | Diversity 3D |
|---|---|---|---|---|---|---|---|---|
| **FLOWR \| SPINDR** | | Base 100 | $0.96 \pm 0.20$ | $1.00 \pm 0.00$ | $0.94 \pm 0.13$ | $0.48 \pm 0.17$ | $0.85 \pm 0.07$ | $0.10 \pm 0.08$ |
| | - | Base 51 | $0.95 \pm 0.22$ | $1.00 \pm 0.00$ | $0.95 \pm 0.12$ | $0.60 \pm 0.25$ | $0.85 \pm 0.07$ | $0.15 \pm 0.13$ |
| | 2 | Cache (AB) | $0.95 \pm 0.21$ | $1.00 \pm 0.00$ | $0.94 \pm 0.13$ | $0.55 \pm 0.24$ | $0.85 \pm 0.07$ | $0.16 \pm 0.11$ |
| | | Cache (TS) | $0.95 \pm 0.21$ | $1.00 \pm 0.00$ | $0.94 \pm 0.13$ | $0.55 \pm 0.22$ | $0.85 \pm 0.07$ | $0.16 \pm 0.11$ |
| | - | Base 34 | $0.93 \pm 0.25$ | $1.00 \pm 0.00$ | $0.95 \pm 0.11$ | $0.55 \pm 0.17$ | $0.86 \pm 0.06$ | $0.15 \pm 0.12$ |
| | 3 | Cache (AB) | $0.94 \pm 0.24$ | $1.00 \pm 0.00$ | $0.94 \pm 0.12$ | $0.54 \pm 0.24$ | $0.85 \pm 0.07$ | $0.17 \pm 0.16$ |
| | | Cache (TS) | $0.94 \pm 0.24$ | $1.00 \pm 0.00$ | $0.94 \pm 0.13$ | $0.51 \pm 0.24$ | $0.85 \pm 0.07$ | $0.16 \pm 0.12$ |
| | - | Base 26 | $0.92 \pm 0.27$ | $1.00 \pm 0.00$ | $0.96 \pm 0.11$ | $0.61 \pm 0.24$ | $0.86 \pm 0.06$ | $0.16 \pm 0.14$ |
| | 4 | Cache (AB) | $0.93 \pm 0.25$ | $1.00 \pm 0.00$ | $0.95 \pm 0.11$ | $0.41 \pm 0.09$ | $0.86 \pm 0.06$ | $0.10 \pm 0.08$ |
| | | Cache (TS) | $0.93 \pm 0.26$ | $1.00 \pm 0.00$ | $0.95 \pm 0.12$ | $0.47 \pm 0.19$ | $0.85 \pm 0.07$ | $0.09 \pm 0.11$ |
| **FLOWR.root \| SPINDR** | | Base 100 | $0.99 \pm 0.11$ | $0.87 \pm 0.28$ | $0.76 \pm 0.30$ | $0.39 \pm 0.22$ | $0.79 \pm 0.13$ | $0.12 \pm 0.10$ |
| | - | Base 51 | $0.98 \pm 0.14$ | $0.88 \pm 0.26$ | $0.79 \pm 0.30$ | $0.41 \pm 0.24$ | $0.80 \pm 0.13$ | $0.13 \pm 0.11$ |
| | 2 | Cache (AB) | $0.98 \pm 0.13$ | $0.88 \pm 0.27$ | $0.78 \pm 0.30$ | $0.39 \pm 0.21$ | $0.79 \pm 0.13$ | $0.12 \pm 0.11$ |
| | | Cache (TS) | $0.98 \pm 0.12$ | $0.88 \pm 0.27$ | $0.78 \pm 0.30$ | $0.42 \pm 0.25$ | $0.79 \pm 0.13$ | $0.14 \pm 0.12$ |
| | - | Base 34 | $0.96 \pm 0.19$ | $0.89 \pm 0.25$ | $0.81 \pm 0.28$ | $0.43 \pm 0.26$ | $0.81 \pm 0.12$ | $0.14 \pm 0.11$ |
| | 3 | Cache (AB) | $0.97 \pm 0.16$ | $0.89 \pm 0.26$ | $0.81 \pm 0.28$ | $0.42 \pm 0.26$ | $0.80 \pm 0.12$ | $0.14 \pm 0.12$ |
| | | Cache (TS) | $0.97 \pm 0.17$ | $0.89 \pm 0.26$ | $0.81 \pm 0.28$ | $0.38 \pm 0.21$ | $0.80 \pm 0.12$ | $0.12 \pm 0.10$ |
| | - | Base 26 | $0.95 \pm 0.22$ | $0.90 \pm 0.24$ | $0.83 \pm 0.27$ | $0.41 \pm 0.26$ | $0.81 \pm 0.12$ | $0.14 \pm 0.11$ |
| | 4 | Cache (AB) | $0.96 \pm 0.19$ | $0.89 \pm 0.25$ | $0.83 \pm 0.27$ | $0.41 \pm 0.25$ | $0.81 \pm 0.12$ | $0.14 \pm 0.11$ |
| | | Cache (TS) | $0.96 \pm 0.19$ | $0.89 \pm 0.20$ | $0.83 \pm 0.27$ | $0.41 \pm 0.26$ | $0.81 \pm 0.12$ | $0.14 \pm 0.11$ |
| **FLOWR.root \| Crossdocked** | | Base 100 | $0.98 \pm 0.13$ | $0.92 \pm 0.19$ | $0.71 \pm 0.30$ | $0.32 \pm 0.18$ | $0.78 \pm 0.11$ | $0.07 \pm 0.08$ |
| | - | Base 51 | $0.98 \pm 0.15$ | $0.98 \pm 0.13$ | $0.75 \pm 0.28$ | $0.33 \pm 0.16$ | $0.80 \pm 0.11$ | $0.07 \pm 0.08$ |
| | 2 | Cache (AB) | $0.98 \pm 0.13$ | $0.93 \pm 0.16$ | $0.76 \pm 0.27$ | $0.33 \pm 0.14$ | $0.79 \pm 0.11$ | $0.06 \pm 0.08$ |
| | | Cache (TS) | $0.98 \pm 0.14$ | $0.93 \pm 0.18$ | $0.76 \pm 0.27$ | $0.33 \pm 0.15$ | $0.79 \pm 0.11$ | $0.06 \pm 0.08$ |
| | - | Base 34 | $0.97 \pm 0.18$ | $0.94 \pm 0.16$ | $0.78 \pm 0.27$ | $0.39 \pm 0.23$ | $0.80 \pm 0.11$ | $0.09 \pm 0.11$ |
| | 3 | Cache (AB) | $0.97 \pm 0.18$ | $0.94 \pm 0.15$ | $0.80 \pm 0.25$ | $0.36 \pm 0.20$ | $0.80 \pm 0.11$ | $0.09 \pm 0.09$ |
| | | Cache (TS) | $0.96 \pm 0.19$ | $0.95 \pm 0.14$ | $0.81 \pm 0.25$ | $0.34 \pm 0.17$ | $0.80 \pm 0.11$ | $0.08 \pm 0.08$ |
| | - | Base 26 | $0.95 \pm 0.22$ | $0.94 \pm 0.16$ | $0.80 \pm 0.26$ | $0.32 \pm 0.15$ | $0.81 \pm 0.11$ | $0.07 \pm 0.08$ |
| | 4 | Cache (AB) | $0.96 \pm 0.19$ | $0.95 \pm 0.14$ | $0.82 \pm 0.24$ | $0.37 \pm 0.17$ | $0.81 \pm 0.11$ | $0.09 \pm 0.09$ |
| | | Cache (TS) | $0.96 \pm 0.19$ | $0.95 \pm 0.14$ | $0.82 \pm 0.24$ | $0.39 \pm 0.18$ | $0.81 \pm 0.11$ | $0.09 \pm 0.11$ |

or higher than the uniform step-reduction baseline. For the Tabasco model, the cached variants consistently maintain higher Validity scores than the base model.

# E    Additional Results: Structure-based Generation

Table 7 provides additional evaluations of the FLOWR and FLOWR.root models, emphasizing Novelty, Uniqueness, and Diversity across varying step budgets. Uniqueness scores are higher for uniform step reduction compared to the cached variants for the SPINDR dataset and we can observe opposite results on the Crossdocked data. The results for 2D and 3D Diversity are inconclusive, as neither caching nor uniform reduction demonstrates a consistent advantage or significant distance across all budgets. In terms of Validity, the cached variants consistently perform on par or superior compared to the uniform step-reduction baselines.

Table 8: Comparison of the FLOWR model and its cached variants with additional property-related quality metrics.

| | D | Mode | SA | QED | Rings | Aromatic R. | HDonors | HAcceptors | LogP | MolWt | Lipinski |
|---|---|---|---|---|---|---|---|---|---|---|---|
| | - | SPINDR Test Set | 0.66 ± 0.12 | 0.49 ± 0.22 | 2.98 ± 1.42 | 1.84 ± 1.31 | 2.62 ± 1.68 | 7.30 ± 4.49 | 0.29 ± 3.48 | 390.43 ± 119.82 | 4.00 ± 1.34 |
| FLOWR \| SPINDR | | Base 100 | 0.66 ± 0.13 | 0.51 ± 0.21 | 3.03 ± 1.42 | 1.75 ± 1.22 | 2.55 ± 1.63 | 7.15 ± 4.50 | 0.49 ± 3.41 | 380.23 ± 118.57 | 4.27 ± 1.10 |
| | 2 | Base 51 | 0.66 ± 0.13 | 0.51 ± 0.21 | 3.03 ± 1.45 | 1.66 ± 1.19 | 2.57 ± 1.63 | 7.12 ± 4.47 | 0.49 ± 3.39 | 379.34 ± 118.36 | 4.27 ± 1.10 |
| | | Cache (AB) | 0.66 ± 0.13 | 0.51 ± 0.21 | 3.03 ± 1.43 | 1.73 ± 1.21 | 2.54 ± 1.64 | 7.16 ± 4.48 | 0.49 ± 3.40 | 380.18 ± 118.59 | 4.27 ± 1.10 |
| | | Cache (TS) | 0.66 ± 0.13 | 0.51 ± 0.21 | 3.02 ± 1.42 | 1.73 ± 1.21 | 2.54 ± 1.63 | 7.16 ± 4.49 | 0.48 ± 3.40 | 380.23 ± 118.45 | 4.26 ± 1.10 |
| | 3 | Base 34 | 0.65 ± 0.13 | 0.51 ± 0.21 | 3.01 ± 1.44 | 1.58 ± 1.17 | 2.58 ± 1.64 | 7.08 ± 4.39 | 0.49 ± 3.34 | 378.03 ± 117.79 | 4.28 ± 1.09 |
| | | Cache (AB) | 0.66 ± 0.13 | 0.51 ± 0.21 | 3.01 ± 1.42 | 1.66 ± 1.19 | 2.55 ± 1.63 | 7.12 ± 4.42 | 0.48 ± 3.36 | 378.85 ± 118.03 | 4.28 ± 1.09 |
| | | Cache (TS) | 0.66 ± 0.13 | 0.51 ± 0.21 | 3.01 ± 1.43 | 1.67 ± 1.20 | 2.55 ± 1.64 | 7.12 ± 4.43 | 0.48 ± 3.38 | 378.99 ± 118.45 | 4.27 ± 1.09 |
| | 4 | Base 26 | 0.64 ± 0.13 | 0.51 ± 0.21 | 2.99 ± 1.44 | 1.50 ± 1.15 | 2.62 ± 1.65 | 7.04 ± 4.36 | 0.48 ± 3.31 | 376.81 ± 117.30 | 4.28 ± 1.10 |
| | | Cache (AB) | 0.65 ± 0.13 | 0.51 ± 0.21 | 3.01 ± 1.45 | 1.59 ± 1.18 | 2.56 ± 1.64 | 7.13 ± 4.42 | 0.46 ± 3.35 | 378.55 ± 118.26 | 4.28 ± 1.10 |
| | | Cache (TS) | 0.65 ± 0.13 | 0.51 ± 0.21 | 2.99 ± 1.44 | 1.60 ± 1.17 | 2.56 ± 1.64 | 7.12 ± 4.40 | 0.46 ± 3.36 | 378.13 ± 117.81 | 4.28 ± 1.10 |
| FLOWR.root \| SPINDR | | Base 100 | 0.67 ± 0.13 | 0.48 ± 0.23 | 3.13 ± 1.42 | 1.96 ± 1.34 | 2.72 ± 2.00 | 7.80 ± 4.82 | 0.04 ± 3.73 | 387.09 ± 120.40 | 4.07 ± 1.25 |
| | 2 | Base 51 | 0.65 ± 0.13 | 0.48 ± 0.22 | 3.19 ± 1.45 | 1.88 ± 1.32 | 2.76 ± 2.02 | 7.84 ± 4.79 | -0.04 ± 3.68 | 386.03 ± 120.41 | 4.07 ± 1.25 |
| | | Cache (AB) | 0.66 ± 0.13 | 0.48 ± 0.22 | 3.12 ± 1.42 | 1.93 ± 1.34 | 2.75 ± 2.01 | 7.80 ± 4.82 | 0.01 ± 3.73 | 388.04 ± 120.82 | 4.06 ± 1.26 |
| | | Cache (TS) | 0.66 ± 0.13 | 0.48 ± 0.22 | 3.13 ± 1.42 | 1.93 ± 1.33 | 2.74 ± 2.01 | 7.83 ± 4.81 | 0.00 ± 3.72 | 387.61 ± 120.64 | 4.07 ± 1.26 |
| | 3 | Base 34 | 0.64 ± 0.14 | 0.48 ± 0.22 | 3.25 ± 1.52 | 1.80 ± 1.31 | 2.78 ± 2.01 | 7.82 ± 4.72 | -0.09 ± 3.63 | 384.23 ± 119.55 | 4.08 ± 1.24 |
| | | Cache (AB) | 0.65 ± 0.13 | 0.48 ± 0.22 | 3.15 ± 1.44 | 1.87 ± 1.32 | 2.76 ± 2.01 | 7.84 ± 4.78 | -0.06 ± 3.68 | 387.03 ± 120.73 | 4.07 ± 1.26 |
| | | Cache (TS) | 0.65 ± 0.13 | 0.47 ± 0.22 | 3.14 ± 1.43 | 1.88 ± 1.32 | 2.78 ± 2.02 | 7.87 ± 4.74 | -0.08 ± 3.67 | 387.22 ± 120.80 | 4.07 ± 1.26 |
| | 4 | Base 26 | 0.64 ± 0.14 | 0.48 ± 0.22 | 3.30 ± 1.56 | 1.74 ± 1.29 | 2.81 ± 2.01 | 7.79 ± 4.71 | -0.11 ± 3.61 | 383.13 ± 119.02 | 4.09 ± 1.24 |
| | | Cache (AB) | 0.64 ± 0.13 | 0.48 ± 0.22 | 3.19 ± 1.46 | 1.82 ± 1.31 | 2.77 ± 2.00 | 7.84 ± 4.71 | -0.10 ± 3.63 | 385.80 ± 120.18 | 4.08 ± 1.25 |
| | | Cache (TS) | 0.65 ± 0.13 | 0.48 ± 0.22 | 3.16 ± 1.45 | 1.85 ± 1.32 | 2.77 ± 1.99 | 7.83 ± 4.69 | -0.08 ± 3.63 | 386.61 ± 120.44 | 4.08 ± 1.25 |
| FLOWR.root \| Crossdocked | - | CD Test Set | 0.76 ± 0.11 | 0.55 ± 0.21 | 2.78 ± 1.38 | 1.75 ± 1.13 | 2.67 ± 2.00 | 5.38 ± 3.28 | 2.02 ± 2.78 | 351.71 ± 132.19 | 4.43 ± 1.03 |
| | | Base 100 | 0.74 ± 0.11 | 0.54 ± 0.19 | 2.97 ± 1.40 | 2.14 ± 1.31 | 2.34 ± 2.07 | 5.62 ± 3.50 | 1.96 ± 2.98 | 352.45 ± 130.97 | 4.48 ± 0.97 |
| | 2 | Base 51 | 0.73 ± 0.12 | 0.53 ± 0.19 | 3.02 ± 1.42 | 2.11 ± 1.31 | 2.35 ± 2.05 | 5.69 ± 3.55 | 1.90 ± 2.97 | 352.19 ± 131.70 | 4.48 ± 0.97 |
| | | Cache (AB) | 0.74 ± 0.12 | 0.53 ± 0.19 | 2.98 ± 1.41 | 2.14 ± 1.32 | 2.33 ± 2.06 | 5.64 ± 3.54 | 1.96 ± 3.00 | 353.19 ± 131.97 | 4.47 ± 0.98 |
| | | Cache (TS) | 0.74 ± 0.12 | 0.53 ± 0.19 | 2.98 ± 1.40 | 2.13 ± 1.32 | 2.34 ± 2.07 | 5.65 ± 3.55 | 1.95 ± 2.99 | 352.98 ± 131.65 | 4.47 ± 0.98 |
| | 3 | Base 34 | 0.73 ± 0.12 | 0.54 ± 0.19 | 3.06 ± 1.45 | 2.06 ± 1.29 | 2.36 ± 2.02 | 5.68 ± 3.52 | 1.85 ± 2.91 | 350.46 ± 130.48 | 4.50 ± 0.95 |
| | | Cache (AB) | 0.73 ± 0.12 | 0.53 ± 0.19 | 2.99 ± 1.41 | 2.09 ± 1.30 | 2.33 ± 2.04 | 5.67 ± 3.50 | 1.94 ± 2.96 | 352.91 ± 131.67 | 4.49 ± 0.96 |
| | | Cache (TS) | 0.73 ± 0.12 | 0.53 ± 0.19 | 2.98 ± 1.41 | 2.09 ± 1.30 | 2.35 ± 2.04 | 5.76 ± 3.54 | 1.86 ± 2.95 | 353.08 ± 131.82 | 4.48 ± 0.97 |
| | 4 | Base 26 | 0.72 ± 0.13 | 0.54 ± 0.19 | 3.09 ± 1.49 | 2.01 ± 1.28 | 2.39 ± 2.02 | 5.65 ± 3.52 | 1.85 ± 2.91 | 350.26 ± 130.91 | 4.49 ± 0.95 |
| | | Cache (AB) | 0.72 ± 0.12 | 0.53 ± 0.19 | 3.00 ± 1.42 | 2.05 ± 1.29 | 2.35 ± 2.01 | 5.66 ± 3.50 | 1.90 ± 2.92 | 352.37 ± 131.15 | 4.49 ± 0.96 |
| | | Cache (TS) | 0.72 ± 0.12 | 0.53 ± 0.19 | 2.99 ± 1.43 | 2.07 ± 1.29 | 2.33 ± 2.03 | 5.69 ± 3.46 | 1.91 ± 2.92 | 353.11 ± 131.21 | 4.49 ± 0.95 |

Table 8 presents an evaluation of the similarity between generated molecules and the properties of both the test set and the base model. For metrics including SA, QED, Rings, MolWt, and Lipinski scores, no substantial differences are observed between caching and uniform reduction. The cached variants demonstrate closer alignment with the 100-step base model for Rings, H-Donors, and H-Acceptors. Although uniform step reduction more closely matches the specific test set value for H-Donors, caching more effectively preserves the distributional characteristics of the FLOWR base model.

