# OpenReview forum: "Predictive Feature Caching for Training-free Acceleration of Molecular Geometry Generation"
_TMLR — Accepted by TMLR_

### Review · Reviewer_zCmj · 2026-02-09

**Summary Of Contributions:**

The paper compares two caching strategies (AB, Taylor) on molecular flow models, and demonstrates that these methods result in roughly 2x throughput without significant performance degradation.

**Audience:**

Yes

**Audience Explanation:**

Accelerating flow model inference is a major question in DL.

**Broader Impact Concerns:**

No issues.

**Claims And Evidence:**

Yes

**Claims Explanation:**

The evaluation considers three recent flow models and a large set of metrics under multiple computation budgets. The experiments are comprehensive and show consistent caching behavior.

**Requested Changes:**

No issues.

---

> ### Author Response · Authors · 2026-03-31
>
> We thank the reviewer for their positive assessment. We are glad that the reviewer found the experiments comprehensive and appreciate the feedback on the relevance of this work to the broader deep learning community.

---

### Review · Reviewer_ZFHQ · 2026-03-14

**Summary Of Contributions:**

The paper proposes a training-free inference acceleration method for molecular geometry generation with flow matching models by caching and forecasting intermediate hidden features across solver steps instead of recomputing the full backbone each time. It adapts existing predictive caching approaches from diffusion models—Taylor-series and Adams–Bashforth forecasting—and applies them to the final block of SE(3)-equivariant molecular generators while arguing that the linear forecasting operations preserve equivariance. Experiments are conducted on three pretrained models (SemlaFlow, Tabasco on GEOM Drugs, and FLOWR on SPINDR) and compared against the original models and uniform step reduction at matched function-evaluation budgets. Results report throughput, validity, energy/strain, and docking metrics, with about 2× speedup at similar generation quality.

**Additional Comments:**

None

**Audience:**

Yes

**Audience Explanation:**

Researchers working on molecular generation and diffusion/flow-based generative models may find the results relevant because the paper explores a practical inference acceleration technique that can reduce the computational cost of sampling from pretrained models. Even though the method is largely an adaptation of predictive caching strategies previously studied in diffusion models, demonstrating that similar ideas can be applied to SE(3)-equivariant molecular generators may still be of interest to practitioners concerned with deployment efficiency and large-scale molecule sampling.

**Broader Impact Concerns:**

The work is a systems-oriented optimization and does not introduce new capabilities. The applications are standard in this area, and the method does not raise additional ethical concerns beyond those typical for molecular generation research.

**Claims And Evidence:**

Yes

**Claims Explanation:**

The experiments show that predictive feature caching can improve throughput relative to the original models and compared with uniform step reduction at matched function-evaluation budgets. The evaluation covers multiple pretrained models (SemlaFlow, Tabasco, FLOWR) and reports standard molecular generation metrics such as validity, strain/energy, docking-related metrics, and throughput, which provides some empirical support for the claimed inference speedups.

However, the evidence is not fully convincing. The method is primarily an adaptation of existing predictive caching approaches from diffusion models, and the paper provides limited analysis of when the approximation fails or how sensitive performance is to forecasting order and caching configuration. The experiments are restricted to a small set of pretrained models and datasets, and there is limited comparison with alternative acceleration strategies used in molecular diffusion/flow models. As a result, while the reported results support the claim that caching can improve inference efficiency in the tested setups, the broader claims about robustness and general applicability are less well supported.

**Requested Changes:**

1. Compare with additional acceleration methods for diffusion/flow models (e.g., improved solvers or distillation), not only uniform step reduction.

2. Analyze when predictive caching degrades performance, including sensitivity to forecasting order and sampling steps.

3. Evaluate different caching locations and test on more models or datasets to demonstrate generalization.

---

> ### Author Response · Authors · 2026-03-31
>
> We sincerely thank the reviewer for their constructive feedback and are pleased that our application of predictive caching to SE(3)-equivariant generators offers a practical approach to accelerating inference in molecular sampling. We also value the reviewer’s suggestions for strengthening our experiments.
>
> &nbsp;
>
> >additional acceleration methods for diffusion/flow models (e.g., improved solvers or distillation), not only uniform step reduction.
>
> In this work, we consider training-free acceleration approaches for molecular generation. Existing methods [1-4] are not training-free and require substantial computational resources. Our work addresses a distinct need by introducing a plug-and-play inference strategy applicable to any pretrained model.
>
> To clarify our selection of baseline, we want to point out that uniform step reduction is currently the standard approach for evaluating the speed-quality tradeoff. SOTA models, including SemlaFlow and FLOWR, characterize this tradeoff by uniform step reduction.
>
> We agree that evaluating performance beyond the Euler solver provides valuable insights. Accordingly, we assess the SemlaFlow model in conjunction with the Heun solver in Appendix B3. Our findings indicate that employing a more advanced solver consistently enhances the base model's performance across multiple metrics. We also demonstrate that combining AB with the Heun solver produces strong results across metrics. These results underscore that our approach is not simply an alternative to improved solvers, but rather an orthogonal framework.
>
> > analysis of [...] how sensitive performance is to forecasting order and caching configuration, analyze when predictive caching degrades performance, including sensitivity to [...] sampling steps
>
> We thank the reviewer for their valuable feedback regarding the sensitivity of our method. Our results in Table 2 for higher caching intervals do show a noticeable degradation in generation quality as the approximation becomes more challenging. We also touch upon possible (model-specific) failure cases in our Limitations section, e.g., how non-smooth trajectories or high-frequency feature variations might make a fixed caching interval less effective.
>
> Appendix B1 addresses the reviewer's inquiry about the sensitivity to the number of steps. We perform an ablation sweep on key quality metrics, varying both the inference steps and the caching intervals. The analysis demonstrates that employing a higher base step count in combination with a larger caching interval generally produces superior results. Nevertheless, at higher intervals, cache staleness may arise. In such cases, a lower base step count with a shorter interval can yield higher validity than a higher base step count with a longer interval.
>
> Moreover, we perform an ablation over the caching order in Appendix B2. We evaluate Taylor and AB, which demonstrate only minor performance differences across caching orders. While AB order 4 was less effective in this scenario, the results indicate that default settings such as $j=2$ or $m=1$ generally yield strong performance.
>
> > Evaluate different caching locations and test on more models or datasets to demonstrate generalization.
>
> In response to the reviewer’s suggestion regarding caching locations, we conducted an ablation study (Appendix B4) to evaluate the effect of caching only the first half of the layers. Interestingly, our results indicate that localized layer caching does not enhance performance; rather, it results in a slight reduction in sampling quality compared to our standard approach, in line with similar observations in [5]. We attribute this performance difference to the increased difficulty and noise sensitivity of predicting intermediate representations within the backbone.
>
> To address the reviewer's request for further evaluation of more models and datasets, we expand our benchmarks to include the FLOWR.root model and the CrossDocked2020 dataset. Our updated results in Tables 1 & 2 demonstrate again that predictive caching at an interval of 2 consistently achieves a nearly lossless speedup across most metrics. Additionally, we confirm that caching outperforms uniform step reduction at lower effective step budgets.
>
> &nbsp;
>
> We thank the reviewer for their constructive feedback. We believe the inclusion of these extended experiments and ablation studies has significantly strengthened the empirical foundation of our work.
>
> &nbsp;
>
> [1] Qu, Jingxiang, et al. "GAGA: Gaussianity-Aware Gaussian Approximation for Efficient 3D Molecular Generation."
>
> [2] Hong, Haokai, et al. "Accelerating 3d molecule generation via jointly geometric optimal transport."
>
> [3] Zhang, Zhilong, et al. "Accelerating 3D Molecule Generative Models with Trajectory Diagnosis."
>
> [4] Cao, Zhonglin, et al. "Efficient molecular conformer generation with so (3)-averaged flow matching and reflow."
>
> [5] Han, Jiaqi, et al. "Adaptive Spectral Feature Forecasting for Diffusion Sampling Acceleration."

---

### Review · Reviewer_fTef · 2026-03-23

**Summary Of Contributions:**

This paper transfers predictive feature caching—previously developed for image/video diffusion models—to SE(3)-equivariant flow-matching models for molecular geometry generation. The core idea is to evaluate the full backbone $v_\theta$ only every $D$ solver steps, and at intermediate steps forecast the last block $F^L$'s output using Taylor-series (TaylorSeer) or Adams–Bashforth (AB) linear multistep schemes applied to previously cached outputs. This reduces the number of full network evaluations from $K$ to approximately $K/D$ without any retraining.

**Key strengths:**
- Targets a practically important bottleneck: large-scale molecular screening requires $10^5$–$10^6$ samples, making inference cost dominant.
- Training-free and drop-in compatible with pretrained models, orthogonal to existing training-based accelerations.
- Thorough evaluation across three backbones (SemlaFlow, Tabasco, FLOWR), two tasks (unconditional and structure-based), and domain-relevant metrics (energy, strain, PoseBusters validity, Vina scores, $W_1$ distances).
- Demonstrates composability with graph compilation and TF32 for up to $7\times$ speedup.

**Key weaknesses:**
- he caching algorithms (TaylorSeer, AB-Cache, last-block forecasting) are directly adopted from the image domain with no algorithmic modification, and the SE(3) equivariance preservation requires no special treatment since it follows from the linearity of the caching operation. The contribution is therefore primarily an empirical transfer—practically useful, but methodologically limited.

**Audience:**

Yes

**Audience Explanation:**

Inference cost is a recognized bottleneck for flow-matching and diffusion-based molecular generators, especially when deployed at scale for virtual screening or de novo design. A training-free, model-agnostic acceleration method that is immediately applicable to existing pretrained models has clear practical relevance. The fact that caching composes with other optimizations (compilation, TF32) for multiplicative gains makes it broadly useful across deployment scenarios.

**Broader Impact Concerns:**

No significant concerns. The method accelerates existing molecular generation models without significantly altering their learned distributions, so it does not introduce new risks beyond those already present in the base models.

**Claims And Evidence:**

Yes

**Claims Explanation:**

The core claims—$2\times$ throughput at matched quality, up to $3\times$ with minimal degradation, and $7\times$ when composed with systems optimizations—are supported by the experimental results. The evaluation is conducted across three distinct pretrained backbones and two generation tasks, with matched NFE baselines (uniform step reduction) as controls.

**Requested Changes:**

1. **Detail the hyperparameter tuning protocol.** Table 3 reports optimal Taylor order $m$ and AB order $j$ per model and caching interval $D$, but describes them only as "determined empirically." Please provide the search ranges, number of trials, selection criterion, and a sensitivity analysis showing how much quality degrades with suboptimal choices (e.g., fixing $j=2$ universally). This is important because the paper's selling point is ease of adoption. Practitioners need to know whether reasonable defaults exist or extensive per-model tuning is required.

2. **Add forecast error analysis.** Plotting $\|\hat{F}(h_t) - F(h_t)\|$ over $t$ across different caching intervals $D$ would help practitioners understand where approximation error concentrates (e.g., early vs. late time regimes) and guide the choice of $D$ for new models.

---

> ### Author Response · Authors · 2026-03-31
>
> We greatly appreciate the constructive and insightful feedback, as well as the recognition of the practical significance of our work. Below, we outline how we have addressed the requested revisions in the updated manuscript to provide a clearer roadmap for practitioners interested in applying predictive feature caching within the molecular domain.
>
> &nbsp;
>
> > Detail the hyperparameter tuning protocol.
>
> We have extended Appendix A to include a detailed description of our search methodology. To determine optimal caching hyperparameters (i.e., caching orders), we conduct a grid search on a small subset of samples, evaluating orders 1-3 for Taylor and 2-4 for AB. Configurations are ranked using core domain metrics, such as Validity or Vina Score. Although orders are tuned per caching interval $k$, our results indicate that the optimal order remains largely consistent across caching intervals for a given model.
>
> Additionally, Appendix B2 presents an experiment comparing the performance of Taylor and AB caching across intervals with varying caching orders. Except for AB order 4, which did not yield satisfactory results in this setting, performance differences between various orders are relatively minor. These findings suggest that while further tuning of the caching order can provide additional performance gains for a specific backbone, practitioners can achieve strong results using default configurations, such as $j=2$ or $m=1$.
>
> > Add forecast error analysis.
>
> We have added this experiment to Appendix B5. By plotting the forecasting error (the difference between the cached output and the ground-truth backbone output) for the SemlaFlow model over the generation trajectory, we can observe that the error typically peaks in the middle section of the trajectory and approaches zero toward the end of the generation process as $t \to 1$. As expected, we can also observe that for larger caching intervals, the error increases over the cached steps.
>
> &nbsp;
>
> We thank the reviewer again for these suggestions. The additions have significantly improved the manuscript's quality, particularly with regard to practical utility.

---

> > ### Comment · Reviewer_fTef · 2026-04-22
> >
> > Thank you for addressing both requested changes. I have no further questions and requests.

---

### Decision · Action_Editor_yQ47 · 2026-05-17

**Recommendation:** Accept as is

**Audience:**

Yes

**Audience Explanation:**

The paper addresses a practically important inference bottleneck for molecular diffusion/flow models. A training-free, plug-and-play acceleration method for pretrained molecular generators should be of interest to researchers working on molecular generation, geometric deep learning, and efficient generative modeling.

**Claims And Evidence:**

Yes

**Claims Explanation:**

The paper provides sufficient empirical evidence that predictive feature caching can accelerate molecular geometry generation while largely preserving sample quality. The evaluation spans multiple SE(3)-equivariant flow-matching backbones, generation settings, and molecular quality metrics, and the revised manuscript further strengthens the evidence through additional ablations on solver choice, caching order, caching location, forecast error, and additional benchmarks.